# Mini-heterochromatin domains constrain the *cis*-regulatory impact of SVA transposons in human brain development and disease

Vivien Horváth [1], Raquel Garza[1], Marie E. Jönsson[1], Pia A. Johansson[1], Anita Adami [1], Georgia Christoforidou[1,2], Ofelia Karlsson[1], Laura Castilla Vallmanya[1], Symela Koutounidou [2], Patricia Gerdes [1], Ninoslav Pandiloski[1,2], Christopher H. Douse [2] & Johan Jakobsson [1] ✉

SVA (SINE (short interspersed nuclear element)−VNTR (variable number of tandem repeats)−*Alu*) retrotransposons remain active in humans and contribute to individual genetic variation. Polymorphic SVA alleles harbor gene regulatory potential and can cause genetic disease. However, how SVA insertions are controlled and functionally impact human disease is unknown. Here we dissect the epigenetic regulation and influence of SVAs in cellular models of X-linked dystonia parkinsonism (XDP), a neurodegenerative disorder caused by an SVA insertion at the *TAF1* locus. We demonstrate that the KRAB zinc finger protein ZNF91 establishes H3K9me3 and DNA methylation over SVAs, including polymorphic alleles, in human neural progenitor cells. The resulting mini-heterochromatin domains attenuate the *cis*-regulatory impact of SVAs. This is critical for XDP pathology; removal of local heterochromatin severely aggravates the XDP molecular phenotype, resulting in increased *TAF1* intron retention and reduced expression. Our results provide unique mechanistic insights into how human polymorphic transposon insertions are recognized and how their regulatory impact is constrained by an innate epigenetic defense system.

More than 50% of the human genome is made up of transposable elements (TEs)[1–3]. Three families of TEs are still active in humans: the autonomous long interspersed nuclear element 1 (LINE-1), the non-autonomous *Alu* short interspersed element and the composite element SVA (SINE (short interspersed nuclear element)−VNTR (variable number of tandem repeats)−*Alu*)[4–9]. The mobilization of these elements represents a notable source of genomic variation in the human population and is the underlying cause of some genetic diseases[10].

SVAs are a class of hominoid-specific TEs. They are nonautonomous, depending on the LINE-1 machinery for retrotransposition,

and consist of a fusion of two TE fragments separated by a VNTR[11]. On the basis of their evolutionary age, SVAs are divided into different subfamilies (A–F), of which SVA-E and SVA-F are human-specific and make up about half of the approximately 3,800 fixed SVAs annotated in the human genome[5,12]. In addition to the annotated SVAs, there are thousands of polymorphic SVA alleles in the human population. Current estimates suggest about one new germline SVA insertion in every 60 births[13]. The individual genetic variation caused by polymorphic SVA insertions is thought to contribute to phenotypic variation in the human population and contribute to, or cause, disease[10,14,15]. However,

[1]Laboratory of Molecular Neurogenetics, Department of Experimental Medical Science, Wallenberg Neuroscience Center and Lund Stem Cell Center, Lund University, Lund, Sweden. [2]Laboratory of Epigenetics and Chromatin Dynamics, Department of Experimental Medical Science, Wallenberg Neuroscience Center and Lund Stem Cell Center, Lund University, Lund, Sweden. ✉e-mail: johan.jakobsson@med.lu.se

**Fig. 1 | Characterization of the XDP NPC model system. a**, Schematic of the generation of XDP NPCs. **b**, Bright-field images of control (CNPC1) and XDP (XNPC1) NPCs (top). Immunocytochemistry (bottom) of SOX2 (green) and Nestin (red) in control and XDP NPCs. Scale bars, 200 μm. **c**, Heatmap of NPC marker gene expression in control ($n = 3$) and XDP ($n = 3$) NPCs measured using RNA-seq. **d**, Schematic of the *TAF1* gene locus. The polymorphic XDP SVA is depicted in red. **e**, PCR analysis of genomic DNA identifying the XDP SVA. **f**, Left, genome browser tracks showing gene expression of the *TAF1* gene and a magnification of intron 32 of *TAF1*, highlighting the characteristic intron retention in XDP NPCs.

The XDP SVA and its direction relative to the *TAF1* gene are depicted in red. Right, quantification of *TAF1* intron 32 retention in control ($n = 12$) and XDP ($n = 12$) NPCs. Bars show the normalized mean expression of the group (adjusted $P$ value (Benjamini–Hochberg correction) as calculated by DESeq2 (Wald test, two-sided)). Error bars show the s.e.m. **g**, Quantification of *TAF1* exon 38 expression in control ($n = 12$) and XDP ($n = 12$) NPCs. Bars show the normalized mean expression of the group (adjusted $P$ value (Benjamini–Hochberg correction) as calculated by DESeq2 (Wald test, two-sided)). Error bars show the s.e.m.

SVAs have been notoriously challenging to study because of their highly repetitive nature and little is known about how polymorphic SVA insertions are regulated by the human genome or how they influence phenotypic traits and disease.

SVAs harbor strong gene regulatory sequences that can function as both transcriptional activators and repressors, influencing the expression of genes in the vicinity of their integration site[14,16–22]. Notably, SVAs appear to be particularly potent as *cis*-regulatory elements in the human brain, where they have been linked to enhancer-like activities[19,22]. In line with this, polymorphic SVAs have been linked to several

genetic neurological disorders[23–28]. The most well-characterized of these is X-linked dystonia parkinsonism (XDP), a recessive adult-onset autosomal genetic neurodegenerative disorder[29–31]. XDP is caused by a germline SVA retrotransposition event in intron 32 of *TAF1*, a gene that encodes TATA box-binding protein-associated factor 1, an essential part of the transcriptional machinery[30,32].

It is unclear how the SVA insertion in *TAF1* leads to XDP pathology. The presence of the SVA has been linked to alternative splicing and reduced transcript levels of *TAF1*, which may be a direct consequence of intron retention of the 32nd intron[30,31,33]. In addition, the XDP SVA

**Table 1 | Description of XDP and control cell lines used in this study**

| Status | Participant | Sample ID | Age at onset (years) | Age at collection (years) | Hexameric repeat length (CCCTCT)$_n$ | Relationship |
|---|---|---|---|---|---|---|
| XDP | 33363.C | XNPC1 | 38 | 44 | 40 | Father of CNPC1 |
| | 33109.2B | XNPC2 | 58 | 72 | | Father of CNPC2 and CNPC3 |
| | 32517.B | XNPC3 | 32 | 35 | 49 | Not related |
| Control | 33362.C | CNPC1 | – | 18 | | Son of XNPC1 |
| | 33114.C | CNPC2 | – | 34 | | Son of XNPC2 |
| | 33113.2I | CNPC3 | – | 42 | | Son of XNPC2 |

ID, identifier. Participants were described by Ito et al.[35].

contains an unstable hexameric repeat that exhibits polymorphic variation associated with age of onset and has the potential to be expanded in somatic tissues. Such expanded hexameric repeats have been speculated to be the source of toxic transcripts or to induce transcriptional interference of *TAF1* (ref. 34).

The example of XDP illustrates the important role of polymorphic SVA insertions in human brain disorders. However, although the SVA insertion is the underlying genetic cause of XDP, the molecular mechanism behind how the SVA interferes with *TAF1* expression is unknown and there is still no mechanistic insight into why certain SVA insertions cause brain disorders. For example, there are hundreds of intronic SVA insertions in the human genome that do not cause disease. How is the human brain protected against the strong regulatory impact of SVAs in these cases and what makes the disease-causing SVA insertion in *TAF1* unique?

In this study, we demonstrate that the DNA-binding KRAB zinc finger protein (KZFP) ZNF91 has a key role in protecting the human genome against the *cis*-regulatory impact of SVAs by establishing a dual layer of repressive epigenetic modifications over SVAs in neural cells, including new polymorphic alleles such as the disease-causing XDP SVA. The resulting mini-heterochromatin domains are characterized by the presence of both DNA methylation and H3K9me3. Notably, the presence of ZNF91-mediated heterochromatin on the polymorphic XDP SVA is highly relevant for XDP pathology, as the removal of this heterochromatin domain aggravates the molecular XDP phenotype, resulting in increased intron retention and reduced *TAF1* expression. In summary, our results provide unique mechanistic insights into how human polymorphic TE insertions are recognized and how their potential regulatory impact in neural cells is minimized by an innate epigenetic defense system based on a KZFP.

## Results

### XDP neural progenitor cells (NPCs) to study the epigenetic regulation of SVAs

To investigate the molecular mechanisms controlling SVAs in human neural cells, including the polymorphic XDP SVA, we established an NPC model system using induced pluripotent stem cell (iPS cell) lines derived from three persons with XDP and three control individuals (Fig. 1a and Table 1)[35]. The XDP SVA carriers presented initially with dystonia at a mean age at onset of $42.6 \pm 13.6$ years, similar to what has previously been reported for persons with XDP ($42.3 \pm 8.3$ years) (Table 1)[30,36]. The controls used were unaffected sons of two of the XDP SVA carriers (Table 1). The six iPS cell lines were converted into stable NPC lines (XDP and control NPCs) that could be extensively expanded or differentiated into different neural cell types[37]. The XDP and control NPCs exhibited NPC morphology and expressed NPC markers such as SOX2 and Nestin, monitored with immunocytochemistry (Fig. 1b and Extended Data Fig. 1a). The expression of NPC markers, as well as the lack of expression of pluripotency markers, was also confirmed by RNA sequencing (RNA-seq; Fig. 1c).

The presence of the XDP SVA insertion, which is ~2.6 kbp long and located in the antisense direction in intron 32 of the *TAF1* gene, was confirmed using PCR (Fig. 1d,e). RNA-seq analysis confirmed that XDP NPCs displayed a characteristic retention of intron 32 of *TAF1* ($P < 0.001$, DESeq2) and lower expression of downstream *TAF1* exons ($P = 0.025$, DESeq2), such as exon 38, when compared to control NPCs (Fig. 1f,g). The retention of intron 32 in XDP NPCs appeared to terminate at the 3′ end of the XDP SVA, although mappability issues with this polymorphic SVA did not allow us to determine exactly where transcription ended (Extended Data Fig. 1b). These observations are similar to those previously described for XDP iPS cell and NPC lines[30].

### SVAs are covered by H3K9me3 in NPCs

TEs, including SVAs, are associated with heterochromatin in somatic tissues, which correlates with their transcriptional silencing and may impact their regulatory potential[38]. We chose to characterize the repressive histone mark H3K9me3, which is linked to heterochromatin, in fetal human forebrain tissue, two XDP NPCs and two control NPCs using CUT&RUN analysis (Fig. 2a). The computational analysis of histone marks on SVAs using CUT&RUN data is challenging because of their repetitive nature. This results in a large proportion of ambiguous reads. To avoid false conclusions because of multimapping artifacts, we used a strict unique mapping approach to investigate individual SVA elements (Fig. 2a). With this bioinformatic approach, it is only

**Fig. 2 | ZNF91 is required for H3K9me3 maintenance at SVAs in NPCs.**
**a**, Schematic of the CUT&RUN approach to profile H3K9me3 at SVAs in NPCs and human fetal forebrain tissue. **b**, Heatmap showing enrichment of H3K9me3 over SVAs in human fetal forebrain tissue. Genomic regions ±10 kbp upstream and downstream of the element are shown. **c**, Heatmap showing H3K9me3 enrichment in NPCs. The genomic regions spanning ±10 kbp upstream and downstream of the element are displayed. **d**, Schematic of the CUT&RUN qPCR approach. Bar plots demonstrating the enrichment of H3K9me3 over the XDP SVA in XDP NPCs ($n = 4$) and the lack of enrichment in control NPCs ($n = 4$; two-tailed *t*-test). Bars show the H3K9me3 enrichment in each replicate and error bars represent the s.d. **e**, Schematic of RNA-seq and snRNA-seq experiments in NPCs and human fetal forebrain tissue. **f**, RNA-seq tracks of *ZNF91* expression in NPCs and fetal forebrain. **g**, Top, uniform manifold approximation and projection

(UMAP) showing characterized cell types. Bottom, UMAP representing *ZNF91* expression in different cell types in the fetal brain. **h**, Schematic of the CRISPRi approach including the lentiviral construct and experimental design. **i**, RNA-seq tracks (left) and quantification (right) of *ZNF91* expression in control CRISPRi and *ZNF91* CRISPRi in control ($n = 4$) and XDP ($n = 4$) NPCs. Bars show the normalized mean expression of the group (adjusted *P* value (Benjamini–Hochberg correction) as calculated by DESeq2 (Wald test, two-sided)). Error bars show the s.e.m. **j**, Heatmap showing H3K9me3 over SVAs in control CRISPRi and *ZNF91* CRISPRi in control and XDP NPCs. **k**, Bar graphs showing the effect of *ZNF91* CRISPRi on H3K9me3 over the XDP SVA in XDP and control NPCs ($n = 4$ in each group; two-tailed *t*-test). Bars show the H3K9me3 enrichment in each replicate and error bars represent the s.d.

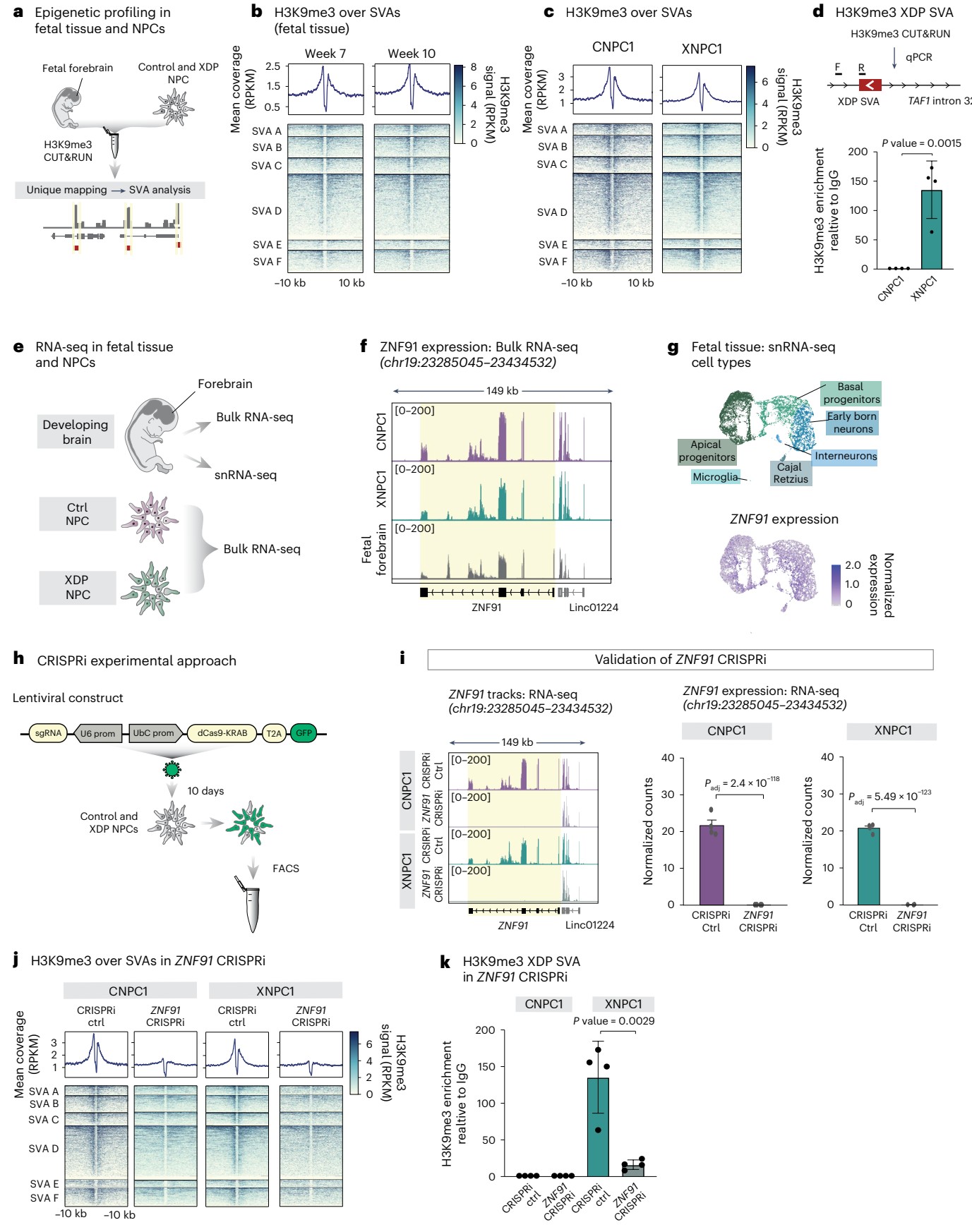

possible to investigate the epigenetic status of the flanking regions of the SVAs where the unique genomic context allows us to discriminate reads without ambiguity and the epigenetic modification can be traced to unique loci in the human genome. The boundaries of nearly all SVAs (>1 kbp in length) of the different subfamilies (A–F), both in the developing human forebrain and in NPCs, were enriched with H3K9me3 (Fig. 2b,c and Extended Data Fig. 2a). However, the genomic context did not enable us to analyze the XDP SVA with this approach. To resolve this issue, we developed a qPCR-based technique, in combination with CUT&RUN (Fig. 2d; see Methods for more details). This analysis, performed with two different primer pairs, revealed a significant enrichment of H3K9me3 at the border of the XDP SVA in XDP NPCs (Fig. 2d and Extended Data Fig. 2b–d).

### H3K9me3 deposition at SVAs is dependent on the KZFP ZNF91

To protect genomic integrity against TE insertions, organisms have evolved cellular defense mechanisms[39,40]. Genes encoding KZFPs have amplified and diversified in mammalian species in response to transposon colonization[41,42]; recent profiling efforts have identified several KZFPs that bind to SVAs, including ZNF91 and ZNF611 (refs. [19,42,43]). We noted that *ZNF91*, in contrast to *ZNF611*, is highly expressed in human fetal forebrain tissue, as well as XDP and control NPC cultures, as monitored by bulk and single nuclei RNA (snRNA)-seq (Fig. 2e–g and Extended Data Fig. 2e)[44]. Thus, we hypothesized that ZNF91 could be a KZFP that binds SVAs and recruits the epigenetic machinery that deposits H3K9me3 at these sites in NPCs.

To investigate a role for ZNF91 in SVA repression in NPCs, we designed a lentiviral clustered regularly interspaced short palindromic repeats (CRISPR) inhibition (CRISPRi) strategy to silence *ZNF91* expression. We targeted two guide RNAs (gRNAs) to a genomic region located next to the *ZNF91* transcription start site (TSS) and coexpressed gRNAs with a KRAB transcriptional repressor domain fused to catalytically dead Cas9 (dCas9) (Fig. 2h). As a control, we used a gRNA targeting *lacZ*, representing a sequence not found in the human genome. The transduction of XDP NPCs and control NPCs resulted in efficient silencing of *ZNF91* expression, monitored with RNA-seq (Fig. 2i and Extended Data Fig. 2f). CUT&RUN analysis of *ZNF91* CRISPRi NPCs (XDP and control NPCs) revealed almost complete loss of H3K9me3 around SVAs (Fig. 2j). This finding was reproduced in one additional XDP NPC line and one additional control NPC line (Extended Data Fig. 2g). CUT&RUN qPCR confirmed that the XDP SVA also lost H3K9me3 in a ZNF91-dependent manner in XDP NPCs (Fig. 2k and Extended Data Fig. 2h). Using a similar CRISPRi strategy, we also confirmed that the H3K9me3 at SVAs, including the XDP SVA, also depends on TRIM28, an epigenetic corepressor protein that is essential for the repressive action of KZFPs, in control and XDP NPCs (Extended Data Fig. 2i–k)[41,45]. Together, these results demonstrate that a ZNF91–TRIM28-dependent mechanism establishes local H3K9me3 heterochromatin over SVAs in human NPCs, including the polymorphic disease-causing XDP SVA.

### SVAs are covered by DNA methylation in human NPCs

In addition to H3K9me3, TE silencing in somatic tissues has been extensively linked to DNA CpG methylation[40,46–49]. To investigate the presence of DNA methylation on SVAs in NPCs, we performed genome-wide methylation profiling using Oxford Nanopore Technologies (ONT)

long-read sequencing (Fig. 3a)[50,51] on one XDP NPC line (XNPC1) and one control NPC line (CNPC1). The long-read DNA methylation analysis revealed that the SVA elements of different subfamilies (A–F), which are GC-rich sequences, were all heavily methylated in human NPCs (Fig. 3b). In addition, the polymorphic XDP SVA was fully covered by DNA methylation (Fig. 3c). Furthermore, we performed Cas9-targeted ONT sequencing over the XDP SVA on the *ZNF91* CRISPRi and control CRISPRi XDP NPCs (Fig. 3d). These results demonstrated that the XDP SVA was fully methylated in both the *ZNF91* CRISPRi and the control CRISPRi XDP NPCs (Fig. 3e). Thus, SVAs in human NPCs, including the XDP SVA, are covered by both DNA methylation and H3K9me3. Our results also indicate that the presence of DNA methylation at SVAs is not dependent on ZNF91 binding or H3K9me3 in this cell type. Notably, the DNA methylation at the XDP SVA did not appear to spread to the surrounding genome, including the *TAF1* TSS, which was located 74.2 kbp upstream of the insertion site (Extended Data Fig. 3a).

### ZNF91 establishes DNA methylation during early development

Because the presence of DNA methylation on SVAs in NPCs did not depend on ZNF91, we wondered how and when DNA methylation on SVAs is established. DNA methylation is reprogrammed during the first few days of early development where global DNA methylation patterns, including those of many TEs, are erased and reinstated[52–56]. During this process, TEs are initially silenced by dynamic epigenetic mechanisms, which are then gradually replaced by other[57], more stable epigenetic mechanisms in somatic cell types, such as NPCs[45,58–61]. We and others have implicated TRIM28–KZFP complexes in this process[45,59,61–65]. Thus, we hypothesized that ZNF91–TRIM28 may be involved in dynamically establishing DNA methylation of SVAs during earlier phases of human embryonic development (Fig. 3f).

To test this hypothesis, we used iPS cells that resemble the epiblast stage of early human development[66], where DNA methylation patterns are more dynamically regulated (Fig. 3f). In contrast, NPCs are somatic cells with a stably methylated genome[49,56,67]. We found clear evidence of dynamic ZNF91-mediated DNA methylation patterning of SVAs when we generated *ZNF91* CRISPRi iPS cells. By performing genome-wide ONT analysis, we found numerous SVAs (n = 39) where DNA methylation was lost upon inhibition of ZNF91 (Fig. 3g). In contrast, the same SVAs were covered by DNA methylation in control iPS cells and NPCs (Fig. 3g). Notably, when we differentiated the *ZNF91* CRISPRi iPS cells into NPCs, we found that these SVAs remained hypomethylated (Fig. 3g). Thus, without *ZNF91* expression, DNA methylation could not be established on these SVAs upon differentiation. For example, an SVA-F element located upstream of the *HORMAD1* gene was fully methylated in control NPCs and iPS cells. The DNA methylation over this SVA was completely lost in *ZNF91* CRISPRi iPS cells and remained absent when the *ZNF91* CRISPRi iPS cells were differentiated into NPCs (Fig. 3h). These results demonstrate that the DNA methylation patterns over some SVAs are dynamic in iPS cells and depend on ZNF91. In addition, ZNF91 is essential for establishing the stable layer of DNA methylation found over these SVAs in NPCs. Thus, cellular context is important for the downstream consequence of ZNF91 binding to SVAs. In early development, ZNF91 mediates the establishment of both H3K9me3 and DNA methylation. On the other hand, only H3K9me3 depends on ZNF91 in somatic cells, whereas DNA methylation is propagated through other mechanisms.

**Fig. 3 | SVAs are covered by DNA methylation in NPCs. a**, Schematic of ONT sequencing experiment to monitor DNA methylation over SVAs. **b**, Methylation coverage over SVAs in control and XDP NPCs. The different SVA families (A–F) are shown. Box plot centers correspond to the median, hinges correspond to the first or third quartile and whiskers stretch from the first or third quartile + 1.5 interquartile range (IQR; n = 1). **c**, Methylation coverage over the XDP SVA in control and XDP NPCs. **d**, Schematic of Cas9-targeted ONT sequencing. **e**, Targeted ONT sequencing in control CRISPRi and *ZNF91* CRISPRi NPCs. The *TAF1* XDP SVA locus is shown. **f**, Schematic of DNA methylation patterns during development in iPS cells and NPCs. *ZNF91* CRISPRi in iPS cells and their conversion to NPCs are also shown. **g**, Violin plot showing DNA methylation over the first quarter (from their TSS) of the differentially expressed SVAs (P value as calculated by Student's t-test (two-sided)). Box plot centers correspond to the median, hinges correspond to the first or third quartile and whiskers stretch from the first or third quartile + 1.5 IQR (wild-type iPS cell n = 1; wild-type NPC, n = 2; iPS cell and iPS cell to NPC ZNF91 KD, n = 1). **h**, DNA methylation pattern over an SVA element near the *HORMAD1* gene.

**a** Oxford nanopore seq. in NPCs

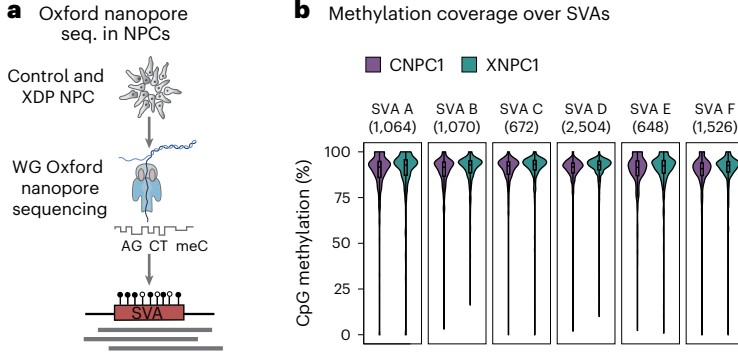

**b** Methylation coverage over SVAs

**c** Methylation coverage over the XDP SVA

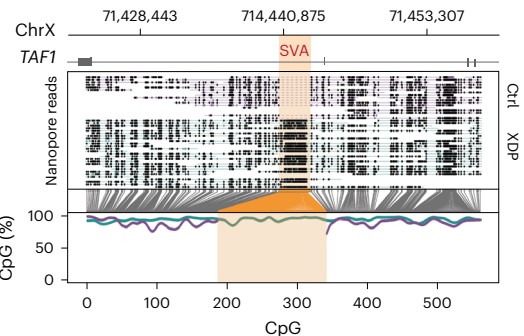

**d** Schematic of Cas9-targeted Oxford nanopore sequencing

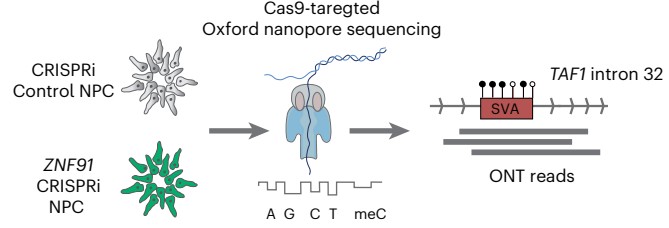

**e** Targeted ONT: *ZNF91* CRISPRi

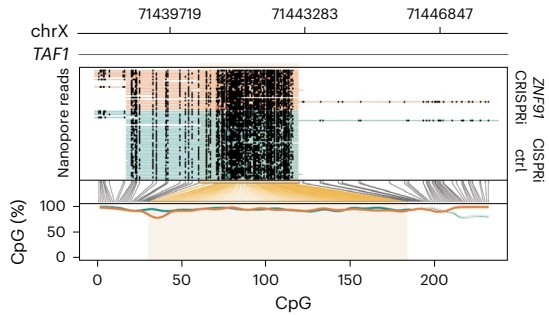

**f** Schematic of experimental design

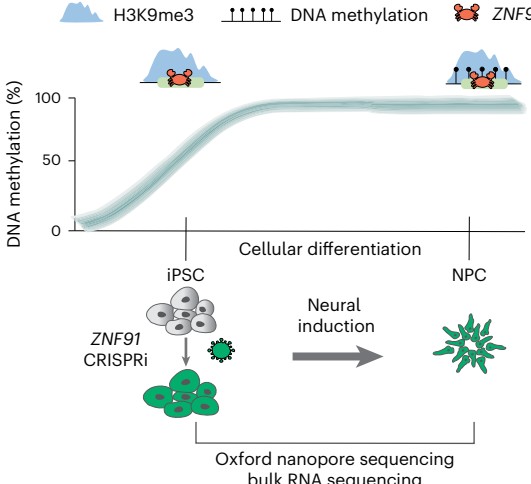

**g** Methylation coverage of DE SVAs

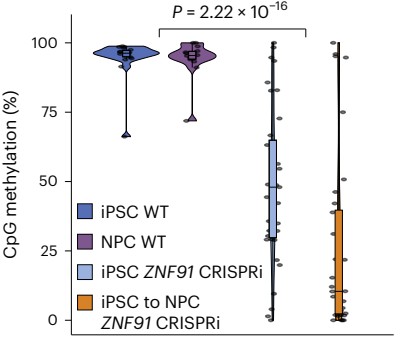

**h** Methylation coverage at the *HORMAD1* locus

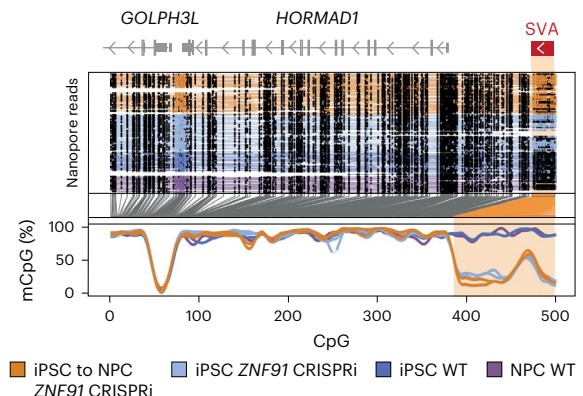

## DNA methylation and H3K9me3 cooperate to silence SVAs

To investigate the role of H3K9me3 and DNA methylation in the transcriptional silencing of SVAs in NPCs, we combined loss-of-function experiments with RNA-seq analysis. To remove H3K9me3, we used the *ZNF91* CRISPRi NPCs. To remove DNA methylation, we deleted DNA methyltransferase 1 (*DNMT1*), which is the gene encoding the enzyme that maintains DNA methylation during cell division[68]. We used a previously described CRISPR-cut approach, resulting in a global loss of DNA methylation, including over SVAs[49,69], as well as a CRISPRi combination strategy targeting the expression of both *DNMT1* and *ZNF91* in XDP and control NPCs (Fig. 4a). Both approaches resulted in a global loss of DNA methylation, as monitored with 5-methylcytosine (m⁵C) immunocytochemistry (Fig. 4b and Extended Data Fig. 3b), and a loss of DNA methylation over the XDP SVA, as demonstrated by targeted ONT long-read sequencing methylation analysis (Fig. 4c). CUT&RUN analysis on NPCs with knockout (KO) of *DNMT1* revealed that loss of DNA methylation did not affect the presence of H3K9me3 at SVAs (Extended Data Fig. 3c).

We used an in-house 2 × 150-bp poly(A)-enriched stranded library preparation for bulk RNA-seq using a reduced fragmentation step to optimize the read length for SVA analysis. Such reads can be uniquely assigned to many SVA loci. We obtained ~40 million reads per sample. To quantify SVA expression, we discarded all ambiguously mapping reads and only quantified those that mapped uniquely to a single location (unique mapping)[70]. We found that *ZNF91* CRISPRi in NPCs, which removed H3K9me3 at SVAs, did not result in activation of SVA expression (Fig. 4d). When *DNMT1* was deleted in NPCs, which removed DNA methylation, we also found only a small number of SVAs transcriptionally upregulated (Fig. 4d). However, in the *ZNF91–DNMT1* double CRISPRi NPCs, where both H3K9me3 and DNA methylation over SVAs were lost, we found a massive transcriptional activation of hundreds of SVAs. To confirm that the SVAs were transcriptionally activated, we performed CUT&RUN analysis for the histone mark H3K4me3, which is associated with active promoters. As the signal of this histone modification spreads to the unique flanking genomic context, this approach allows for an accurate identification of transcriptionally active individual transposon loci[44]. This analysis confirmed that most full-length SVAs gained H3K4me3 upon *ZNF91–DNMT1* double CRISPRi (Fig. 4d,e and Extended Data Fig. 3d).

We also analyzed the expression of SVAs in the iPS cell–NPC conversion experiments (Fig. 3f). RNA-seq revealed that the SVAs that lost DNA methylation after *ZNF91* deletion in iPS cells (*ZNF91* CRISPRi iPS cells; Fig. 3g) were also transcriptionally upregulated (Fig. 4f). This contrasted with *ZNF91* CRISPRi NPCs, where the same SVA elements were not upregulated upon inhibition of ZNF91 (Fig. 4f). When we analyzed the *ZNF91* CRISPRi iPS cells that were differentiated to NPCs, we found that the SVAs were expressed in these NPCs (Fig. 4f). These SVAs were also found to be upregulated upon *ZNF91–DNMT1* double CRISPRi in NPCs (Fig. 4f). One example was an SVA-E element located upstream of the *RNF24* gene (Fig. 4g). This SVA was transcriptionally silent in control iPS cells and NPCs. In *ZNF91* CRISPRi iPS cells, we detected a robust activation of the expression of this SVA-E element, also shown by the presence of the H3K4me3 peak, which correlated with the loss of DNA

methylation. When the *ZNF91* CRISPRi iPS cells were differentiated to NPCs, the SVA remained expressed; this also correlated with a lack of DNA methylation. Thus, the loss of DNA methylation patterns over SVAs in iPS cells upon *ZNF91* CRISPRi correlated with the transcriptional activation of SVAs, including when these cells were differentiated to NPCs. These experiments demonstrate that ZNF91 dynamically represses the expression of at least some SVAs in iPS cells and is essential for establishing stable transcriptional repression of these SVAs.

## DNA methylation and H3K9me3 restrict SVA *cis*-regulation

SVAs carry regulatory sequences that can mediate *cis*-acting transcriptional effects on the surrounding genome[16–21]. We, therefore, investigated whether the ZNF91-mediated heterochromatin domains found over SVAs in NPCs influenced this activity. When investigating transcriptional changes of genes monitored by RNA-seq upon removal of H3K9me3 (*ZNF91* CRISPRi), removal of DNA methylation (*DNMT1* KO) or removal of both repressive marks (*ZNF91–DNMT1* double CRISPRi), we only found profound effects on nearby gene expression in the *ZNF91–DNMT1* double CRISPRi NPCs. The expression of genes located in the vicinity of an SVA element were significantly increased upon *ZNF91–DNMT1* double CRISPRi but not when deleting only one of the factors (Fig. 5a). This effect could be detected when the SVA was located up to 50 kbp from the TSS but was stronger when the SVA was closer to the TSS (Fig. 5a).

Notably, the dynamics of the SVA-mediated influence on gene expression was distinct between different loci. Most genes in the vicinity of an SVA were completely unaffected by *ZNF91* deletion or *DNMT1* deletion alone but transcriptionally upregulated when both factors were removed (Fig. 5a). Thus, in most instances, the presence of one of the heterochromatin marks was sufficient to protect flanking genomic regions from the regulatory impact of SVAs. However, we also found examples where both marks were needed to block the regulatory impact of SVAs. For example, the expression of *HORMAD1* was upregulated because of the activation of an upstream SVA-F element acting as an alternative promoter in both *ZNF91* CRISPRi and *DNMT1* KO NPCs (Fig. 5b). When both *ZNF91* and *DNMT1* were inhibited, *HORMAD1* expression was even more strongly activated, suggesting a cooperative mode of action (Fig. 5b). This demonstrates that both epigenetic marks are necessary at some loci to block the regulatory impact of SVAs.

## TAF1 XDP phenotype exacerbated by loss of heterochromatin

We next used the XDP NPCs to investigate whether the presence of H3K9me3 and DNA methylation over the XDP SVA has any impact on *TAF1* expression. Removing H3K9me3 alone (*ZNF91* CRISPRi) affected neither intron retention in the *TAF1* loci in the XDP NPCs nor exon 38 expression of the *TAF1* gene (Fig. 5c,d). When we investigated the *TAF1* loci in XDP NPCs that lacked DNA methylation (*DNMT1* KO), retention of intron 32 was significantly increased and exon 38 expression was reduced (Fig. 5c,d). Removing both DNA methylation and H3K9me3 (*ZNF91–DNMT1* double CRISPRi) had an even stronger effect on *TAF1* expression, including a considerable increase in intron 32 retention of *TAF1* and lower expression of exon 38 in XDP NPCs (Fig. 5c,d). We also observed the appearance of antisense transcripts originating from

**Fig. 4 | DNA methylation and H3K9me3 cooperate to silence SVAs in NPCs.** **a**, Schematic of CRISPR-cut and double CRISPRi experiment in NPCs. **b**, m⁵C immunostaining showing the global loss of DNA methylation upon *DNMT1* CRISPRi. Scale bars, 100 μm. **c**, DNA methylation coverage over the XDP SVA in control CRISPR and *DNMT1* CRISPR-cut conditions. **d**, Left, heatmap) showing upregulated SVAs in *ZNF91–DNMT1* double CRISPRi. The same SVAs are also shown in *ZNF91* CRISPRi and *DNMT1* KO experiments. Right, box plot showing SVA expression in *ZNF91* CRISPRi, *DNMT1* KO and *ZNF91–DNMT1* double CRISPRi. Box plot centers correspond to the median, hinges correspond to the first or third quartile and whiskers stretch from the first or third quartile + 1.5 IQR; outliers are indicated by points (*n* = 4, except for *ZNF91–DNMT1* double CRISPRi,

where *n* = 3). **e**, Heatmap showing H3K4me3 enrichment in control NPCs and NPCs with double KD of *ZNF91* and *DNMT1*. Genomic regions spanning ±10 kbp upstream and downstream of the element are shown. **f**, Left, heatmap showing the expression level of differentially methylated SVAs. Right, box plot showing the expression of differentially expressed SVAs. Box plot centers correspond to the median, hinges correspond to the first or third quartile and whiskers stretch from the first or third quartile + 1.5 IQR; outliers are indicated by points (*n* = 4, except for *ZNF91–DNMT1* double CRISPRi, where *n* = 3). **g**, Genome browser tracks showing gene expression (top left), H3K4me3 (bottom left) and DNA methylation (right) pattern over an SVA element near the *RNF24* gene.

**a** Schematic of CRISPR cut and double CRISPRi

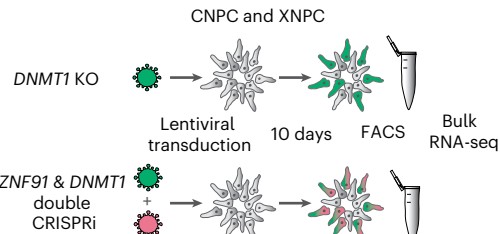

**b** 5 mC immunostaining

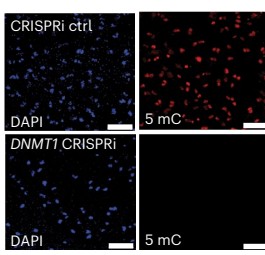

**c** Targeted ONT: *DNMT1* CRISPR cut

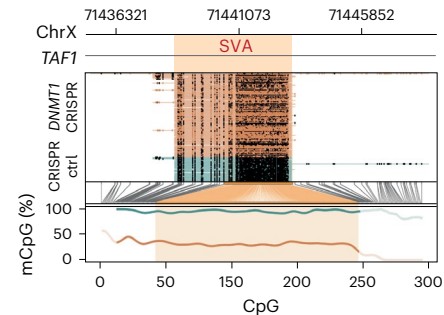

**d** SVA expression (NPCs): RNA-seq

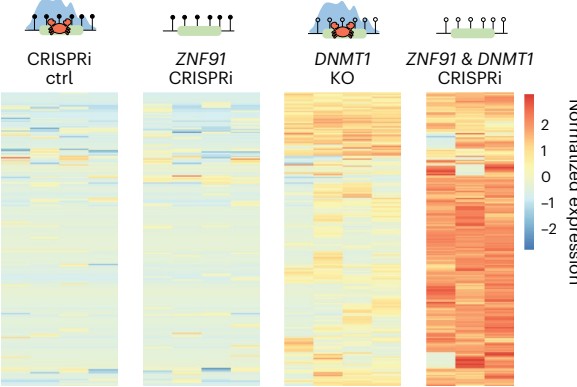

**e** H3K4me3 over DE SVAs

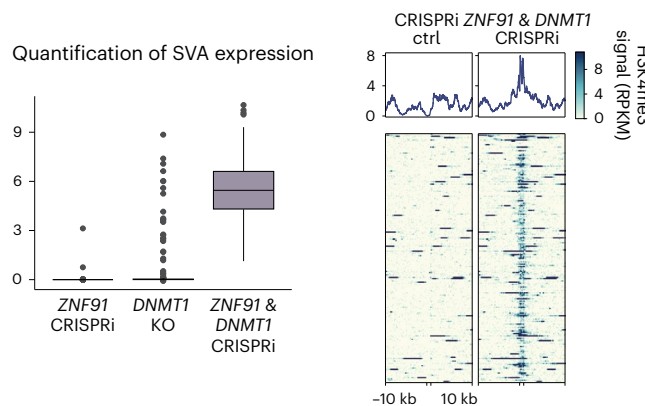

**f** Differentially methylated SVA expression: RNA-seq

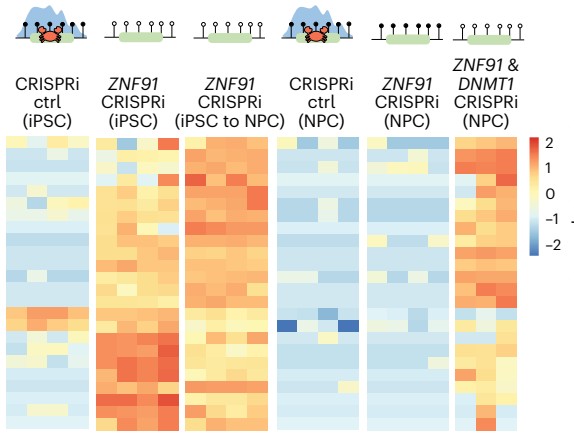

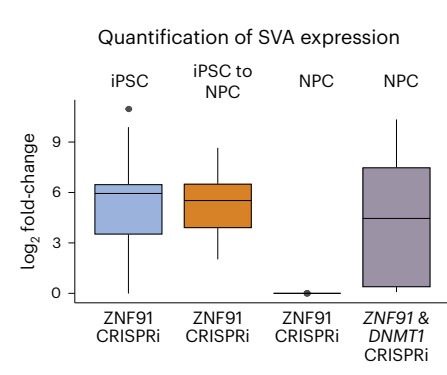

**g** Changes in expression and methylation coverage at the *RNF24* locus

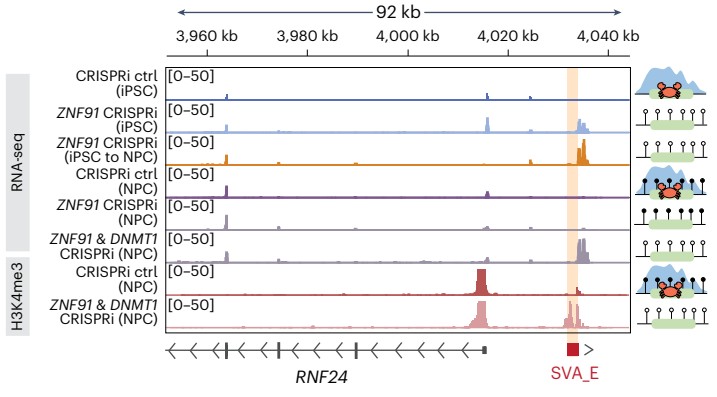

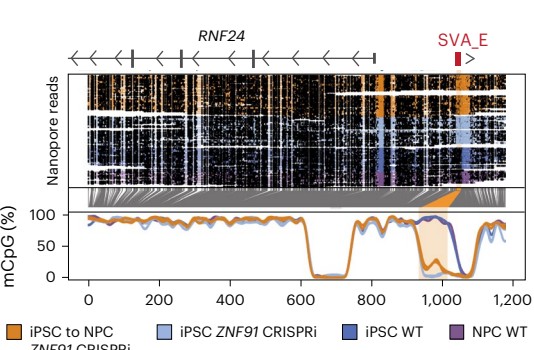

**a** SVA effect on nearby gene expression

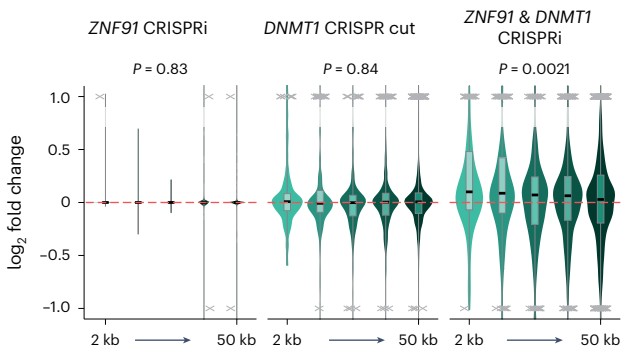

**b** *HORMAD1* expression

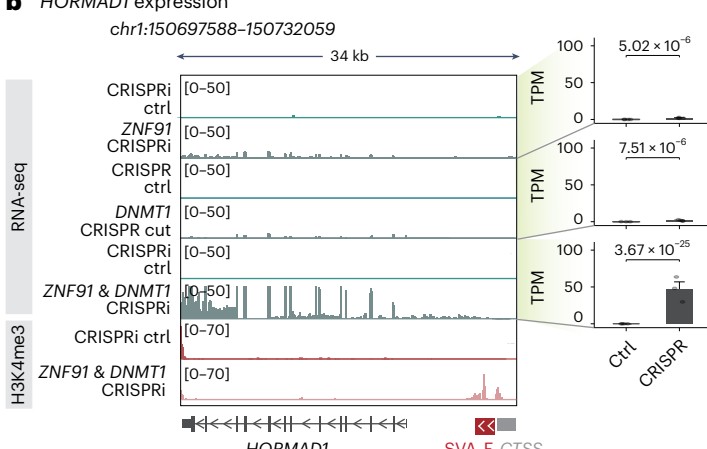

**c** *TAF1* intron 32 expression

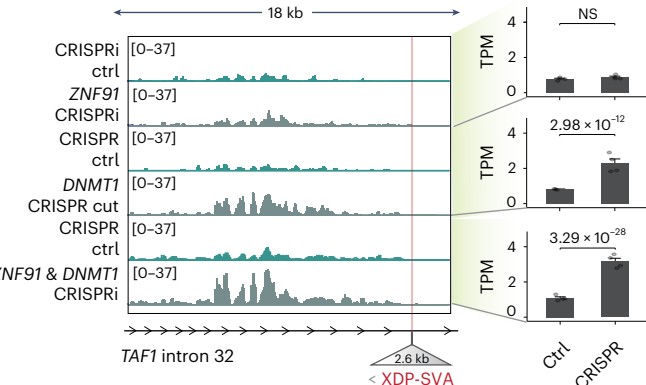

**d** *TAF1* exon 38 expression

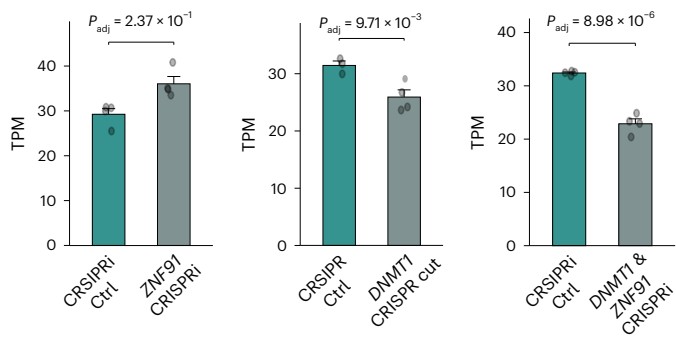

**Fig. 5 | SVAs have a regulatory influence on nearby genes when heterochromatin marks are lost. a**, Violin plot showing the effect of SVAs on nearby gene expression (2–50 kbp) in *ZNF91* CRISPRi, *DNMT1* KO and *ZNF91*–*DNMT1* double CRISPRi. *P* value was calculated by a one-way analysis of variance (two-sided). Box plot centers correspond to the median, hinges correspond to the first or third quartile and whiskers extend until the minimum and maximum values; outliers and data points outside of the plot's axes are indicated by crosses (*ZNF91* CRISPRi, *n* = 6; *DNMT1* CRISPR-cut, *n* = 2; *ZNF91*–*DNMT1* double CRISPRi, *n* = 2). **b**, Left, genome browser tracks showing *HORMAD1* expression and H3K4me3 in *ZNF91* CRISPRi, *DNMT1* KO and *ZNF91*–*DNMT1* double CRISPRi. Right, bar plots showing normalized mean expression of *HORMAD1* in *ZNF91* CRISPRi, *DNMT1* KO and *ZNF91*–*DNMT1* double CRISPRi (*n* = 4). The adjusted

*P* value (Benjamini–Hochberg correction) was calculated by DESeq2 (Wald test, two-sided). Error bars show the s.e.m. **c**, Left, genome browser tracks showing *TAF1* intron 32 expression in *ZNF91* CRISPRi, *DNMT1* KO and *ZNF91*–*DNMT1* double CRISPRi. Right, bar plots showing normalized mean expression of *TAF1* intron 32 in *ZNF91* CRISPRi, *DNMT1* KO and *ZNF91*–*DNMT1* double CRISPRi (*n* = 4). The XDP SVA is depicted in red. The adjusted *P* value (Benjamini–Hochberg correction) was calculated by DESeq2 (Wald test, two-sided). Error bars show the s.e.m. **d**, Bar plots showing normalized mean expression of *TAF1* exon 38 in *ZNF91* CRISPRi, *DNMT1* CRISPR-cut and *ZNF91*–*DNMT1* double CRISPRi (*n* = 4). The adjusted *P* value (Benjamini–Hochberg correction) was calculated by DESeq2 (Wald test, two-sided). Error bars show the s.e.m.

the XDP SVA in the *ZNF91*–*DNMT1* double CRISPRi NPCs (Extended Data Fig. 1b). This suggests that the XDP SVA becomes transcriptionally active in *ZNF91*–*DNMT1* double CRISPRi NPCs, a finding that was corroborated by the presence of H3K4me3 at the boundaries of the XDP SVA in NPCs, as monitored by our qPCR CUT&RUN approach (Extended Data Fig. 3e). Thus, the loss of both DNA methylation and H3K9me3 exacerbated molecular pathology in XDP NPCs. These data demonstrate that the regulatory impact of the polymorphic XDP SVA is negatively influenced by the presence of a local mini-heterochromatin domain. When this heterochromatin domain is lost, the *cis*-regulatory effect of the XDP SVA is strongly and significantly enhanced.

**Polymorphic SVA insertions are silenced by ZNF91**

To investigate whether the local heterochromatin observed over the polymorphic XDP SVA represented a unique event or was a general effect, we extended our analysis to other polymorphic SVAs in the genomes of two of the individuals in this study. We took advantage of the whole-genome ONT long-read sequencing data from the XNPC1

and CNPC1 lines and used the transposons from long DNA reads (TLDR) pipeline to identify non-reference SVA insertions (Fig. 6a)[51]. We identified 12 high-confidence polymorphic insertions of the SVA-E and SVA-F subfamilies (average length of 1.3 kbp), eight of which were shared between the two genomes (Fig. 6b and Supplementary Table 1). Notably, several of these polymorphic SVAs, which represent recent TE insertions into the germline of these two individuals, displayed clear hallmarks of local heterochromatinization, including the presence of DNA methylation and H3K9me3 (Supplementary Table 1).

For example, we found a polymorphic SVA insertion present only in XNPC1 in an intron of *SLC12A6*, which is a gene encoding a potassium–chloride transporter linked to neurological disorders[71,72]. This SVA insertion site displayed H3K9me3 at its boundaries and was fully covered by DNA methylation (Fig. 6c). *ZNF91* CRISPRi led to loss of H3K9me3 over this SVA, while *ZNF91*–*DNMT1* double CRISPRi in XNPC1 led to its transcriptional activation, resulting in the expression of an antisense readthrough transcript extending into the *SLC12A6* gene (Fig. 6c). Another example was a polymorphic SVA insertion shared

**a** Schematic of polyTE annotation

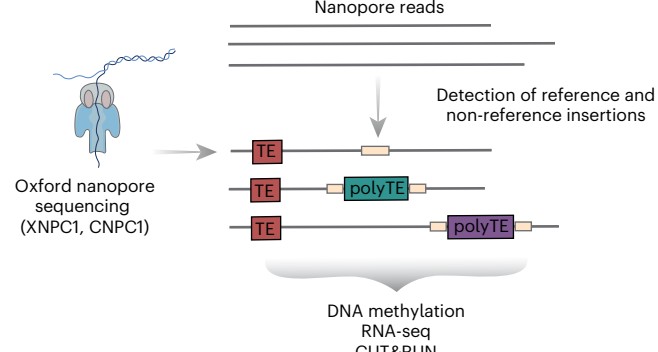

**b** Annotated polymorphic SVAs

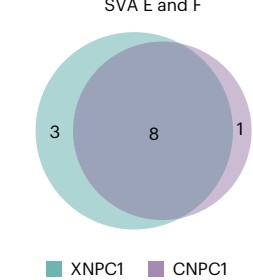

**c** Polymorphic SVA insertion in *SLC12A6* locus

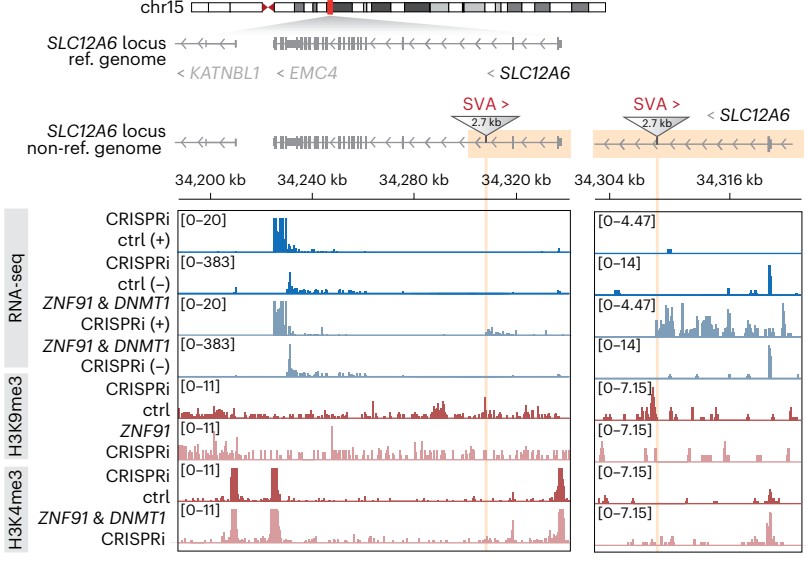

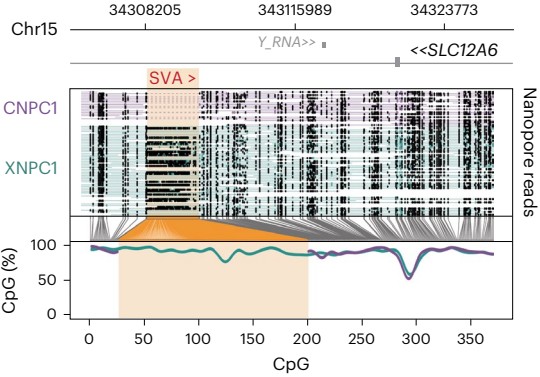

**d** Polymorphic SVA insertion near *GABPA* locus

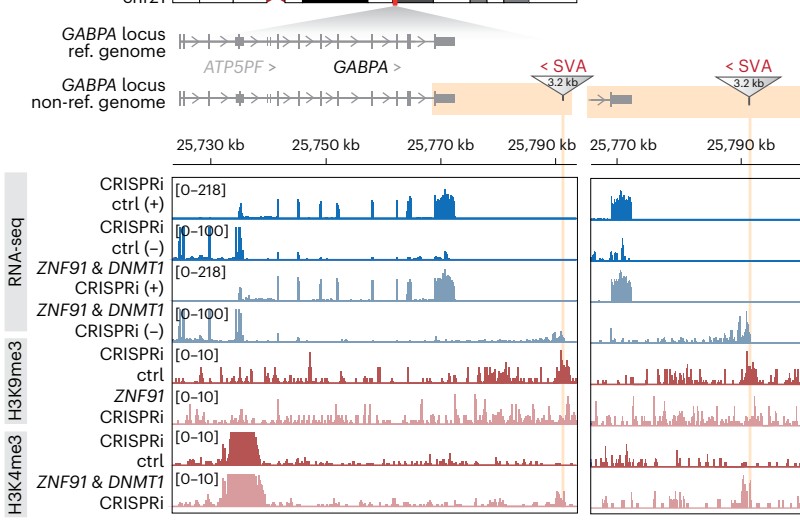

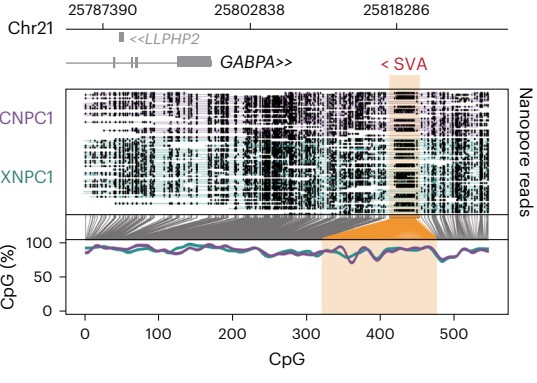

**Fig. 6 | Polymorphic SVA insertions are repressed by DNA methylation and H3K9me3. a**, Schematic of ONT sequencing and annotation of polymorphic SVAs. **b**, Venn diagram showing annotated polymorphic SVA insertions. **c**, Left, genome browser tracks showing gene expression, H3K9me3 and H3K4me3 over a polymorphic SVA insertion and the nearby gene *SLC12A6*. Right, ONT sequencing data showing DNA methylation over the annotated polymorphic insertion. **d**, Left, genome browser tracks showing gene expression, H3K9me3 and H3K4me3 over a polymorphic SVA insertion and the nearby gene *GABPA*. Right, ONT sequencing data showing DNA methylation over the annotated polymorphic insertion.

between CNPC1 and XNPC1 downstream of *GABPA*, which is a gene encoding a DNA-binding protein involved in mitochondrial function[73,74]. Similarly, the SVA insertion resulted in the accumulation of H3K9me3 and DNA methylation; *ZNF91* CRISPRi and *ZNF91–DNMT1* double CRISPRi resulted in the loss of H3K9me3 and transcriptional activation of the SVA, respectively, generating a readthrough antisense transcript (Fig. 6d).

These data confirm that recent, polymorphic germline SVA insertions are recognized by ZNF91 in human NPCs and are covered by a dual layer of repressive epigenetic marks. Loss of this local heterochromatin domain results in transcriptional activation of these elements and the formation of novel transcripts, which are likely to have a regulatory impact on nearby genes. In these cases, this was illustrated by the production of polymorphic antisense transcripts to *SLC12A6* and *GABPA*.

## Discussion

Our data support a model in which the KZNF ZNF91 binds to SVAs in early human development and throughout brain development, modeled herein using iPS cells and NPCs. ZNF91 binding results in the establishment of local heterochromatin over SVAs, characterized by DNA methylation and H3K9me3, in a TRIM28-dependent manner. In early development (represented herein by iPS cells), both modifications are dependent on the binding of ZNF91 to SVAs. In later phases of development, after epigenetic reprogramming of the genome (represented herein by NPCs), only H3K9me3 depends on ZNF91, whereas DNA methylation over SVAs is propagated by DNMT1. These two repressive chromatin modifications work together to limit the influence of SVAs on the host genome. They prevent expression of the SVA elements and restrict the *cis*-acting influence of SVAs on the surrounding regions in neural cells. It is worth noting that this mechanism not only involves evolutionarily older SVA insertions that are fixed in the human population but also includes recent polymorphic germline SVA insertions, including the disease-causing XDP SVA. There are limitations to these conclusions. Although the iPS cell–NPC model system used in this project recapitulates some aspects of human brain development, it is a simplified model of early brain development. To validate some of the findings in this study, we relied on human fetal tissue. However, such analysis does not provide much mechanistic insight and we cannot exclude the possibility that some of our observations were in vitro artifacts. In addition, the iPS cell–NPC system does not allow us to investigate differences in the epigenetic silencing of SVAs over time and in different cell types. It will be interesting to address such questions using more complex model systems, such as cerebral organoids.

SVAs carry regulatory sequences with the potential to exert strong *cis*-acting influences on gene regulatory networks[16–21,23]. The ZNF91-mediated mini-heterochromatin domains prevented this *cis*-acting influence in the NPC model system. Our data demonstrate that, for most SVA loci, only one of the heterochromatin marks was necessary to silence SVA expression and to prevent its regulatory influence on nearby gene expression. However, there are examples where the cooperation of the two mechanisms appeared necessary. The most striking example was the *HORMAD1* locus, where an upstream SVA could act as an alternative promoter[43]. The regulatory effect of this SVA was activated when H3K9me3 and DNA methylation were removed individually, demonstrating the need for both marks to prevent the *cis*-acting influence of this SVA. Removal of both marks resulted in a massive activation of *HORMAD1* expression, indicating that dual removal has a synergistic effect. It is not yet understood why some SVA loci, such as that upstream of *HORMAD1*, require both mechanisms for their control. However, it is likely that the transcriptional and epigenetic state of the integration sites is important, as well as structural variants within SVAs. For example, it is known that the VNTR region of SVAs is highly variable and has expanded recently in human evolution[51,75]. It is also evident that ZNF91 heterochromatin domains are not able to prevent the regulatory influence of all new SVA germline insertions.

SVA insertions on the sense strand within genes are less abundant than expected by chance, suggesting that such SVA insertions are selected against[11,76]. In addition, there are a growing number of polymorphic SVA insertions linked to genetic disorders, with XDP being the best-characterized example.

The SVA insertion linked to XDP is in intron 32 of the essential gene *TAF1*; the molecular phenotype includes intron retention and reduced *TAF1* expression[30,31]. Our data demonstrate that ZNF91 binds to the XDP SVA and establishes a polymorphic mini-heterochromatin domain. The epigenetic status of the XDP SVA is highly relevant for XDP pathology, as this layer of heterochromatin protects against the gene regulatory impact of the SVA. When DNA methylation and H3K9me3 are lost, intron retention is greatly increased, *TAF1* expression levels are further reduced and the XDP SVA is transcriptionally activated. Thus, while ZNF91 can limit the impact of the XDP SVA insertion, it cannot entirely remove the regulatory impact over the *TAF1* gene. This explains why the XDP SVA causes disease while most other SVA insertions are inert. However, we still do not understand why ZNF91 is unable to fully block the *cis*-regulatory impact of the XDP SVA.

Our data are limited to cell culture models; the epigenetic status of the XDP SVA in human brain tissue has not been extensively studied. However, long-read ONT data obtained from postmortem brain tissue of one individual suggested that SVAs were highly methylated in the adult brain, although the methylation level appeared to be slightly lower in polymorphic SVA insertions[51]. In addition, the presence of DNA methylation over the XDP SVA was investigated in postmortem brain tissue from a patient[77]. This analysis showed that, although the XDP SVA was methylated in brain tissue, the methylation levels in the brain were reduced compared to blood samples from the same individual. It is worth noting that DNA methylation patterns in the human brain change with age[78–80]. It will be interesting to investigate whether DNA methylation over the XDP SVA is stable in the human brain or lost with age. Such phenomena could explain the late-onset phenotype of XDP, where a gradual increase in the loss of *TAF1* function ultimately results in cellular dysfunction. Such a scenario would also open new therapeutic possibilities where restoration of DNA methylation on the XDP SVA could block or reverse the pathology. However, further analysis using larger cohorts of postmortem XDP tissue is required to investigate this possibility.

KZFPs have been implicated in an evolutionary arms race with TEs, where expansions and modifications of genes encoding KZFPs limit the activity of newly emerged transposon classes[42]. This event is followed by mutations in the TEs to avoid repression in an ongoing cycle. One such example is ZNF91, which appeared in the last common ancestor of humans and Old World monkeys and underwent a series of structural changes about 8–12 million years ago that enabled it to bind to SVA elements[42,43]. However, the presence of the SVA-binding ZNF91 in hominoids has not prevented the expansion of SVAs in their genomes. On the contrary, SVAs are highly active in the human germline, providing a substantial source of genomic variation in the population[5,13,14,23,81–83]. Although ZNF91 is not able to completely prevent new SVA germline insertions, sustained expression during brain development greatly limits the *cis*-regulatory impact of these insertions. Thus, it appears that, in this case, ZNF91 may facilitate the expansion of SVA insertions by limiting their gene regulatory impact on the human genome. Our data are consistent with a model where KZFPs are not only TE repressors but also facilitators of inert germline transposition events, thereby fueling genome complexity and evolution.

Our results demonstrate how a KZFP prevents the regulatory impact of TEs in human neural cells but it is still not known whether the ZNF91–SVA partnership represents a unique event or whether these results can be extrapolated to other TE families and lineages. In humans, there are three active TE classes: *Alu* elements, LINE-1s and SVAs. The relationship between LINE-1s and SVAs is of special interest because SVA retrotransposition depends on coexpression of the LINE-1

machinery. We and others previously found that LINE-1s are controlled by DNA methylation in human NPCs and their transcriptional activation is recognized and silenced by the human silencing hub (HUSH) complex by a mechanism that is independent of KZFPs and TRIM28 (ref. [84]). Thus, in human brain development, LINE-1 activity appears to be controlled by fundamentally different mechanisms to SVAs. This suggests that the control of TE activity in the human brain is not only multilayered but also highly specialized. Our data are limited to cell models of early development and neural cells, where *ZNF91* is particularly highly expressed. We do not know how SVAs are controlled in other human tissues, although high DNA methylation of SVAs has been shown in the heart and liver[51]. In addition, our data indicate that there are additional KZFPs controlling SVAs in early human development (see one example in Extended Data Fig. 4a and Supplementary Table 1). ZNF91 deletion in iPS cells activated only a fraction of SVAs, whereas all others remained silenced. It is likely that these SVAs contain binding sites for additional KZFPs, such as ZNF611, that cooperate to control SVAs in early development[19,43]. ZNF91 may then have a unique role in neural tissues. It is the only KZFP that binds SVAs and protects against *cis*-acting mechanisms from regulatory sequences in SVAs that are highly active in neural cells.

In summary, our results provide unique mechanistic insights into an epigenetic defense system, based on a KZFP, active against the regulatory impact of SVA transposons in the human brain. On one hand, this system protects the genome from any negative impact of SVAs, with SVA insertions resulting in genetic disease only in very rare instances, exemplified herein by XDP. On the other hand, this system has likely contributed to the expansion of SVAs in our genomes, maximizing the potential for TEs to contribute to increased genome complexity and suggesting that SVAs likely had an important role in primate brain evolution.

## Online content

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

## Methods

All iPS cells used in this study were previously described[30]. All research using patient-derived iPS cells was performed according to national legislation. Before experimental use, all cell lines were confirmed to be free of mycoplasma.

### iPS cells

We used iPS cell lines derived from three persons with XDP and three healthy individuals from WiCell (Table 1). iPS cells were maintained on Biolaminin 521-coated (0.7 mg cm⁻²; Biolamina) Nunc multidishes in iPS medium (StemMACS iPS-Brew XF and 0.5% penicillin–streptomycin (Gibco)). Cells were passaged 1:3 every 2–3 days. Briefly, cells were rinsed once with Dulbecco's PBS (DPBS; Gibco) and dissociated using Accutase (Gibco) at 37 °C for 5 min. Following incubation, Accutase was carefully aspirated from the well and the cells were washed off from the dish using washing medium (9.5 ml of DMEM/F-12 (Gibco) and 0.5 ml of KO serum replacement (Gibco)). The cells were then centrifuged at 400g for 5 min and resuspended in iPS-Brew medium supplemented with 10 mM Y27632 ROCK inhibitor (Miltenyi Biotech) for expansion. The medium was changed daily[85].

### NPCs

The NPCs were generated from iPS cells from the three persons with XDP and three unaffected individuals (Table 1). The neural induction was performed as previously described[86]. The NPCs were cultured in DMEM/F-12 (Thermo Fisher Scientific) supplemented with glutamine (2 mM; Sigma), penicillin–streptomycin (1×; Gibco), N2 supplement (1×; Thermo Fisher Scientific), B27 (0.05×; Thermo Fisher Scientific), epidermal growth factor (EGF) and fibroblast growth factor 2 (FGF2) (both 10 ng ml⁻¹; Thermo Fisher Scientific). Additionally, 10 mM Y27632 ROCK inhibitor (Miltenyi) was used. Cells were grown on Nunc multidishes or in T25 flasks precoated with poly-L-ornithine (15 µg ml⁻¹; Sigma) and laminin (2 µg ml⁻¹; Sigma). Cells were passaged every 2–3 days using TryplE express enzyme (Gibco) and trypsin inhibitor (Gibco).

### Immunocytochemistry

First, 24-well Nunc plates were precoated with poly-L-ornithine (15 µg ml⁻¹; Sigma) and laminin (2 µg ml⁻¹; Sigma). Approximately 50,000 cells were plated in the wells and were allowed to expand until they reached 70–80% confluency. At this point, cells were washed three times with DPBS (Gibco), fixed with 4% paraformaldehyde (Merck Millipore) solution for 15 min at room temperature and washed again three times with DPBS. Fixed cells were stored in DPBS at 4 °C for a maximum of 1 month until staining and imaging.

For blocking, cells were incubated for 1 h with 5% normal donkey serum (NDS) in TKBPS (PBS with potassium phosphate (KBPS) with 0.25% Triton X-100; Fisher Scientific). Subsequently, they were incubated overnight at 4 °C with the corresponding primary antibody (m⁵C from Active Motif, cat. no. 39649, lot 02617020, 1:250; SOX2 from R&D Systems, AF2018, 1:100; Nestin from Abcam, AB176571, 1:100). For a negative control, cells were incubated overnight with TKPBS + 5% NDS. After overnight incubation, cells were washed two times for 5 min in TKBPS, followed by 5 min in TKBPS with NDS. Next, they were incubated at room temperature for 2 h with the secondary antibody (donkey anti-rabbit Alexa fluor 647 from Jackson Lab, 1:200; donkey anti-goat cy3 from Jackson Lab, 1:200) and for 5 min with DAPI (1:1000; Sigma Aldrich) as a nuclear counterstain. This was followed by two 5-min washes with KPBS; then, cells were stored in PBS until imaging.

### m⁵C staining

As described by Jönsson et al.[49], cells stained for m⁵C were pretreated with 0.9% Triton in PBS for 15 min, followed by 2 N HCl for 15 min and then 10 mM Tris-HCl pH 8 for 10 min before incubation with the primary antibody. Cells were imaged using a fluorescence microscope (Leica).

### RNA-seq

Total RNA was isolated using the RNeasy Mini Kit (Quiagen) with on-column DNAse treatment following the manufacturer's instructions. The isolated RNA was used for qPCR (see Methods, 'CUT&RUN qRT–PCR') and RNA-seq. RNA-seq was performed using four biological replicates. Libraries for RNA-seq were generated using Illumina TruSeq Stranded mRNA library prep kit (poly(A) selection), optimized for long fragments and sequenced on a Novaseq6000 (paired end, ~250 bp), yielding an average of 46 million reads. The reads were mapped to the human reference genome (hg38) using STAR aligner (version 2.7.8a)[87] and gene quantification was performed using featureCounts (Subread package version 1.6.3; hg38 GENCODE version 38), setting the flags '-p' and '-s 2' for paired-end and reversely stranded reads, respectively (TruSeq)[88].

To quantify TE expression, reads were remapped using STAR aligner and discarded if mapped to more than one location ('--outFilterMultimapNmax 1'). A maximum of 0.03 mismatches per base were allowed ('--outFilterMismatchNoverLmax 0.03'). featureCounts (Subread package version 1.6.3) using the hg38 RepeatMasker annotation 'parsed to filter out low complexity and simple repeats, rRNA, scRNA, snRNA, srpRNA and tRNA' was used to quantify reads[89].

BigWig files for genome browser tracks were generated using bamCoverage (deepTools version 2.5.4), set to '-normalizeUsingRPKM' and '-filterRNAstrand' to split signal between strands. Visualization was performed in the Integrative Genome Browser (IGV)[90]. Matrices for deepTools heatmaps were generated including only SVAs longer than 1 kbp (grouped by subfamily; individual BED files), using computeMatrix scale-regions setting '-regionBodyLength' to 1 kbp and flanking regions ('-a' and '-b') to 10 kbp. Heatmaps were generated using plotHeatmap (version 3.5.1). Profile plots were generated the same way, ungrouping the SVAs (using a single BED file as input) before the matrix computation.

Normalization of counts to visualize the expression of different features on bar plots (genes, *TAF1* intron 32 or *TAF1* exon 38) was performed as TPM (transcript per million); the length of the feature was used to calculate an approximate TPM value. Statistical tests, however, were performed using DESeq2, which normalizes using the median of ratios[91]. Intron 32 and exon 38 of the *TAF1* gene were added as part of the gene count matrix DESeq2 model correcting for sequencing batch and the individuals' diagnosis (XDP or control) whenever appropriate.

### CUT&RUN

H3K9me3 CUT&RUN analysis was performed on CNPC1, CNPC2, XNPC1 and XNPC3 in both *ZNF91* CRISPRi and *TRIM28* CRISPRi, as well as control CRISPRi (*lacZ*). H3K4me3 CUT&RUN was performed on CNPC1 and XNPC1 in *ZNF91–DNMT1* double CRISPRi conditions. We followed the protocol described by Skene and Henikoff[92]. Briefly, 300,000 cells were washed twice (20 mM HEPES pH 7.5, 150 mM NaCl, 0.5 mM spermidine and 1× Roche cOmplete protease inhibitors) and attached to ten ConA-coated magnetic beads (Bangs Laboratories) that were preactivated in binding buffer (20 mM HEPES pH 7.9, 10 mM KCl, 1 mM CaCl₂ and 1 mM MnCl₂). Bead-bound cells were resuspended in 50 ml of buffer (20 mM HEPES pH 7.5, 0.15 M NaCl, 0.5 mM spermidine, 1× Roche cOmplete protease inhibitors, 0.02% w/v digitonin and 2 mM EDTA) containing primary antibody (rabbit anti-H3K9me3, Abcam ab8898, RRID: AB_306848; rabbit anti-H3K4me3, Active Motif cat. no. 39159, RRID: AB_2555751; goat anti-rabbit IgG, Abcam ab97047, RRID: AB_10681025) at 1:50 dilution and incubated at 4 °C overnight with gentle shaking. Beads were washed thoroughly with digitonin buffer (20 mM HEPES pH 7.5, 150 mM NaCl, 0.5 mM spermidine, 1× Roche cOmplete protease inhibitors and 0.02% digitonin). After the final wash, pA-MNase (a generous gift from S. Henikoff) was added in digitonin buffer and incubated with the cells at 4 °C for 1 h. Bead-bound cells were washed twice, resuspended in 100 ml of digitonin buffer and chilled to 0–2 °C. Genome cleavage was stimulated by adding

2 mM CaCl$_2$ at 0 °C for 30 min. The reaction was quenched by adding 100 ml of 2× stop buffer (0.35 M NaCl, 20 mM EDTA, 4 mM EGTA, 0.02% digitonin, 50 ng ml$^{-1}$ glycogen, 50 ng ml$^{-1}$ Rnase A and 10 fg ml$^{-1}$ yeast spike-in DNA (a generous gift from S. Henikoff)) and vortexing. After 10-min incubation at 37 °C to release genomic fragments, cells and beads were pelleted by centrifugation (16,000$g$, 5 min, 4 °C) and fragments from the supernatant were purified. Illumina sequencing libraries were prepared using the Hyperprep kit (KAPA) with unique dual-indexed adaptors (KAPA), pooled and sequenced on a Nextseq500 instrument (Illumina). Paired-end reads (2 × 75) were aligned to the human and yeast genomes (hg38 and R64-1-1, respectively) using Bowtie 2 ('--local --very-sensitive --local --no-mixed --no-discordant --phred33 -I 10 -X 700') and converted to BAM files using SAMtools[93]. Normalized BigWig coverage tracks were made using bamCoverage (deepTools)[94], with a scaling factor accounting for the number of reads arising from the spike-in yeast DNA (10$^4$ per aligned yeast read number). Tracks were displayed in IGV.

### CRISPR approaches

**CRISPRi.** To silence the transcription of *ZNF91* and *TRIM28*, we used dCas9 fused to the transcriptional repressor KRAB[95]. Single-guide sequences were designed to recognize DNA regions just downstream of the TSS, according to the Genetic Perturbation Platform (GPP) Portal (Broad Institute). *ZNF91* sgRNA: GAGTTTCCAGGTCTCGACTT (no protospacer adjacent motif (PAM)). The guides were inserted into a dCas9-KRAB-T2A-GFP lentiviral backbone containing both the guide RNA under the U6 promoter and dCas9–KRAB and GFP under the Ubiquitin C promoter (pLV hU6-sgRNA hUbC-dCas9-KRAB-T2a-GFP, a gift from Charles Gersbach, Addgene plasmid 71237, RRID: Addgene_71237). The guides were inserted into the backbone using annealed oligos and the BsmBI cloning site. Lentiviruses were produced as described below, yielding titers of 10$^8$–10$^9$ TU per ml, which was determined using qRT–PCR. Control virus with a gRNA sequence absent from the human genome (*lacZ*) was also produced and used in all experiments. All lentiviral vectors were used with a multiplicity of infection (MOI) of 2.5 unless stated differently. GFP cells were isolated by fluorescence-activated cell sorting (FACS; FACSAria, BD sciences) on day 10 at 10 °C (reanalysis showed >97% purity) and pelleted at 400$g$ for 5 min, snap-frozen on dry ice and stored at −80 °C until RNA isolation. All groups were performed in four biological replicates unless indicated differently. Knockdown (KD) efficiency was validated using RNA-seq.

***DNMT1* CRISPR-cut.** LV.gRNA.Cas9-GFP vectors were used to target *DNMT1* (ref. 69) or *lacZ* (control) as previously described[49]. Lentiviral vectors were produced as described previously and had a titer of 10$^8$–10$^9$ TU per ml, which was determined using qRT–PCR. Human NPCs were transduced with an MOI of 10–15, allowed to expand for 10 days and sorted by FACS as described previously.

***DNMT1–ZNF91* double CRISPRi.** We used a double-transduction method to perform a double CRISPRi of *DNMT1* and *ZNF91*. We transduced *ZNF91* with the previously mentioned dCas9-KRAB-T2A-GFP lentivirus containing the dCas9 protein and a GFP. At the same time, the cells were transduced with lentivirus containing the pLV.U6BsmBI. EFS-NS.H2b-RFPW lentiviral backbone with the gRNA for *DNMT1* and mCherry as a marker but without dCas9 to knock down *DNMT1*. *DNMT1* sgRNA: TGCTGAAGCCTCCGAGATGC (no PAM). Double-positive (mCherry and GFP) cells were sorted by FACS as previously described and stored at −80 °C until RNA extraction.

### Lentiviral vector production

Lentiviral vectors were produced according to Zufferey et al.[96]. Briefly, HEK293T cells were grown to a confluency of 70–90% on the day of transfection for lentiviral production. We used third-generation packaging and envelop vectors (pMDL, psRev and pMD2G), together with polyethyleneimine (PEI; Polysciences PN 23966, in DPBS (Gibco)). The lentivirus was harvested 2 days after transfection. The supernatant was then collected, filtered and centrifuged at 25,000$g$ for 1.5 h at 4 °C. The supernatant was removed from the tubes and the virus was resuspended in PBS and left at 4 °C. The resulting lentivirus was aliquoted and stored at −80 °C.

### CUT&RUN qRT–PCR

To identify whether the XDP SVA was surrounded by an H3K9me3 or H3K4me3 mark, we designed a qPCR approach. For the H3K9me3 analysis, we used CNPC1 and XNPC1 control CRISPRi and *ZNF91* CRISPRi cells, four replicates each. For the H3K4me3 analysis, we used CNPC1 and XNPC1 control CRISPRi and *ZNF91–DNMT1* double CRISPRi cells, one replicate each. Briefly, two primer pairs were designed on the 5' region of the XDP SVA: one in the flanking region and one in the SVA (Fig. 2d). As a positive control, primers were designed for a genomic region (hg38, chromosome 5 (chr5):141253464–141255143) known to be covered by H3K9me3. The CUT&RUN library was used as a template for amplification. The qRT–PCR was performed with SYBR Green I Master (Roche) on a LightCycler 480 (Roche). The primer pairs used were as follows:

XDP SVA forward 1 (5′–3′): GAATGGTATATGTTTAGTTTTACA
XDP SVA reverse 1 (5′–3′): CATGACCCTGCCAAATCCCCCT
XDP SVA forward 2 (5′–3′): TAGTTTTACAAGACACGGCACTATT
XDP SVA reverse 2 (5′–3′): CAGGGTCCTCTGCCTAGGAAAACC
Positive control forward (5′–3′): AAATGGGAATTAAAATCAGTGAGG
Positive control reverse (5′–3′): TTGACATATCATTAAGGGGGCA

### ONT sequencing

**Whole-genome ONT sequencing.** DNA was extracted from frozen pellets using the Nanobind HMW DNA Extraction kit (PacBio) following the manufacturer's instructions. The final product was eluted in 100 μl of elution buffer provided in the kit. DNA concentration and quality were measured using Nanodrop and Qubit from the top, middle and bottom of each tube. Only DNA with a quality of 260/280 1.8–2.0 and 260/230 2.0–2.2 was further processed. Whole-genome sequencing on samples XNPC1 and CNPC1 was performed at the SciLife lab in Uppsala using SQK-LSK109 Ligation Sequencing kit (ONT) and FLO-PRO002 PromethION Flow Cell R9 Version on a PromethION (ONT).

**Cas9-targeted ONT sequencing.** To target the XDP locus, we used the Cas9 sequencing (SQK-CS9109) kit following the manufacturer's instructions (ONT). To enrich for the fragment of interest, four previously described gRNAs were used[97]. Briefly, two guides were designed upstream and two were designed downstream of the XDP SVA insertion. The excision using these guides resulted in a 5.5-kbp product, including the XDP SVA (2.6 kbp). A total of 5 μg of DNA was used. Samples were sequenced on a MinION Mk1Mc using flow cell R9.4.1 (ONT). One flow cell per sample was used. For cas9-targeted enrichment, we obtained 20,770 reads from three samples (XNPC3 control CRISPRi, 7,805 reads; XNPC3 *DNMT1* KO, 7,581 reads; XNPC3 *ZNF91* CRISPRi, 4,384 reads; SAMtools view '-c BAM'). The proportion of reads that mapped to the target was 1.25% on average (control CRISPRi, 52 reads; *DNMT1* KO, 171 reads; *ZNF91* CRISPRi, 36 reads; primary alignments over *TAF1* intron 32 only; SAMtools view '-c -F 260 BAM chrX:71424238–71457170'). XNPC3 control CRISPRi, *ZNF91* CRISPRi and *DNMT1* KO samples were sequenced using the targeted approach.

FASTQ files were indexed using the nanopolish index (version 0.13.3) on default parameters[98]. To build an index of the XDP genome (with the XDP SVA insertion), a consensus of the SVA sequences (two enrichment methods: PCR and CRISPR) as reported by Reyes et al.[97] (https://github.com/nanopol/xdp_sva/blob/main/) was created using EMBOSS cons (version 6.6.0.0) (http://emboss.open-bio.org/) resulting in a 2,638-bp sequence.

A *TAF1* FASTA file was generated using grep '-w *TAF1*' from hg38 GENCODE version 38 and BEDTools getfasta (version 2.30.0)[99].

The SVA consensus sequence and the *TAF1* sequence were then aligned using ClustalW2 (version 2.1). We observed that the breaking point between the sequence extracted by Reyes et al. (2022) and the reference genome's sequence of *TAF1* occurred at nucleotide chrX:71,440,502 (ref. [97]).

The FASTA file of chrX was then chopped using BEDTools getfasta using the following breaking points:

chrX171440502

chrX71440503156040895

The three sequences (chrX:1–71440502, the SVA sequence and chrX:71440503–156040895) were then stitched (concatenated) together. A new genome FASTA file was created by concatenating all hg38 chromosome FASTA files (except for chrX) and the XDP chrX FASTA file. A Minimap2 (version 2.24) index was then created using the ONT preset ('-x map-ont')[100]. Mapping of the reads was performed using Minimap2 (version 2.24) using the ONT preset ('-a -x map-ont') with the XDP genome index. BAM files were sorted and indexed using SAMtools (version 1.16.1).

To characterize the XDP SVA hexameric region, reads mapping to the insertion were extracted using samtools view for chrX:71440502–71443153. Consensus sequences of the XDP SVA for each cell line were produced using SAMtools consensus, including insertions and deletions. Hexamer length was defined upon manual curation of their sequence and confirmed using Noise-Cancelling Repeat Finder (https://github.com/makovalab-psu/NoiseCancellingRepeatFinder) using the samples' consensus sequence as the input FASTA and CCCTCT as the motif ('--positionalevents --maxnoise 20% --minlength 0 --stats=events').

Polymorphic insertions were identified using TLDR (version 1.2.2), using GRCh38.p13 as the reference genome ('-r') and a TE library ('-e') including the consensus sequences for TE subfamilies: L1Ta, L1preTa, L1PA2, SVA A–F and HERVK (sequences provided by TLDR developers)[51]. Insertions were required to have an UnmapCover of at least 80% (percentage of insertion with TE sequence), a sequence similarity (TEMatch) of at least 80% to the TE consensus sequence and a minimum of three reads supporting it (span-reads). SVAs were considered for further analysis if their length was greater than 1 kbp.

The local consensus sequences of the polymorphic insertions (as the output from TLDR) were introduced to the reference genome. A custom script (add_polymorphic_insertions_fa.py) was used to read the TLDR output table, sort the polymorphic insertions from the end to the start of each chromosome and perform the following operations for each of the chromosomes:

1. Read its FASTA file (chr.fa).
2. Extract its sequence before and after the insertion using BED-Tools getfasta ('-fi chr.fa -bed coordinates.bed'), where coordinates.bed included two coordinates: the first spanning from the beginning of the chromosome to the start of the insertion (as reported by TLDR) and the second spanning from the end of the insertion (as reported by TLDR) to the end of the chromosome.
3. Concatenate the following sequences, before rewriting the chromosome's FASTA with the output:

   a. The chromosome before the insertion
   b. The sequence of the polymorphic insertion
   c. The chromosome after the insertion

This process was repeated for each polymorphic insertion, introducing them in order from the end to the start of each chromosome. Similarly, an updated gene annotation GTF file (to fit the coordinates including all polymorphic insertions) was created using a custom script following the same logic (add_polymorphic_insertions.py).

The reads were remapped to the custom genome using an indexed version of the output FASTA (output from add_polymorphic_insertions_fa.py) using Minimap2 (indexing using '-x map-ont'; mapping using '-a -x map-ont').

Methylation for each of the regions of interest was called using Nanopolish call-methylation (version 0.13.3) with the raw reads ('-r'), the alignment files to the custom genome ('-b') and the custom genome's FASTA file as a reference ('-g'). Databases for Methylartist were produced using Methylartist db-nanopolish (version 1.2.2), using the methylation call files as input (default parameters). Specific loci were visualized using Methylartist locus (version 1.2.2)[50].

**Reporting summary**

Further information on research design is available in the Nature Portfolio Reporting Summary linked to this article.

## Data availability

All processed sequencing data have been deposited and are available at GSE245093. This paper includes analyses of existing, publicly available data. The accession numbers for these datasets are GSE224747 (3′ snRNA-seq, bulk RNA-seq and H3K9me3 CUT&RUN of human fetal forebrain tissue) and GSE242143 (H3K9me3 CUT&RUN from the *DNMT1* KO NPCs). Source data are provided with this paper.

## Code availability

All original code has been deposited on GitHub (https://github.com/raquelgarza/XDP_Horvath_2023).

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

## Acknowledgements

We would like to thank F. Jacobs, D. Trono, C. Bragg and A. Alessi for comments on the manuscript and their support throughout this project. We also thank J. Johansson, M. Persson Vejgården, A. Hammarberg and U. Jarl for their technical assistance. We are grateful to all members of the Jakobsson lab. The work was supported by grants from the Collaborative Center for X-Linked Dystonia-Parkinsonism (J.J. and C.H.D.), the Swedish Research Council (2018-02694 to J.J. and 2021-03494 to C.H.D.), the Swedish Brain Foundation (FO2019-0098 to J.J.), Cancerfonden (190326 to J.J.), Barncancerfonden (PR2017-0053 to J.J.), the Swedish Society for Medical Research (S19-0100 to C.H.D.), and the Swedish Government Initiative for Strategic Research Areas (MultiPark & StemTherapy).

## Author contributions

All the authors took part in designing the study and interpreting the data. V.H. and J.J. conceived the study. V.H., M.E.J., P.A.J., A.A., G.C., O.K., L.C.V., S.K. and C.H.D. performed the experimental research. R.G. and N.P. performed the bioinformatic analyses. C.H.D. and P.G. contributed expertise. V.H., C.D. and J.J. wrote the manuscript, and all authors reviewed the final version.

## Funding

## Competing interests

The authors declare no competing interests.

## Additional information

**Extended data** is available for this paper at https://doi.org/10.1038/s41594-024-01320-8.

**Correspondence and requests for materials** should be addressed to Johan Jakobsson.

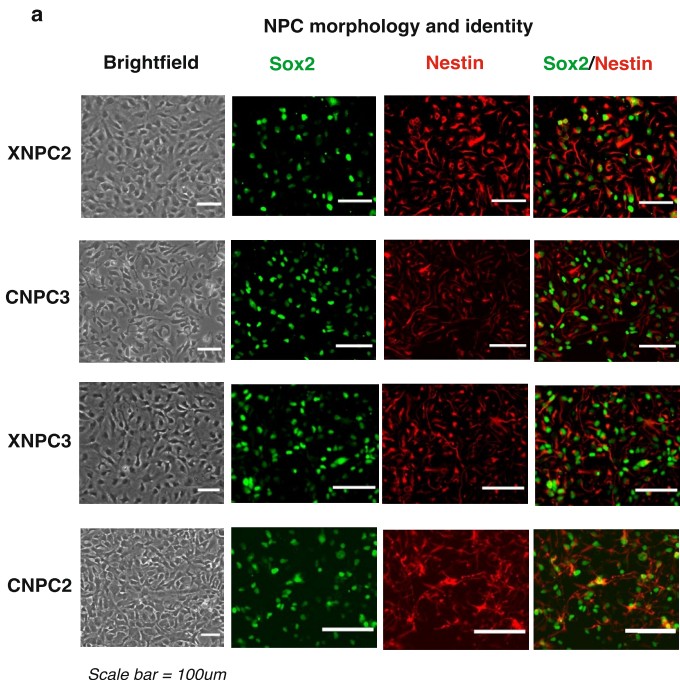

**a**

**NPC morphology and identity**

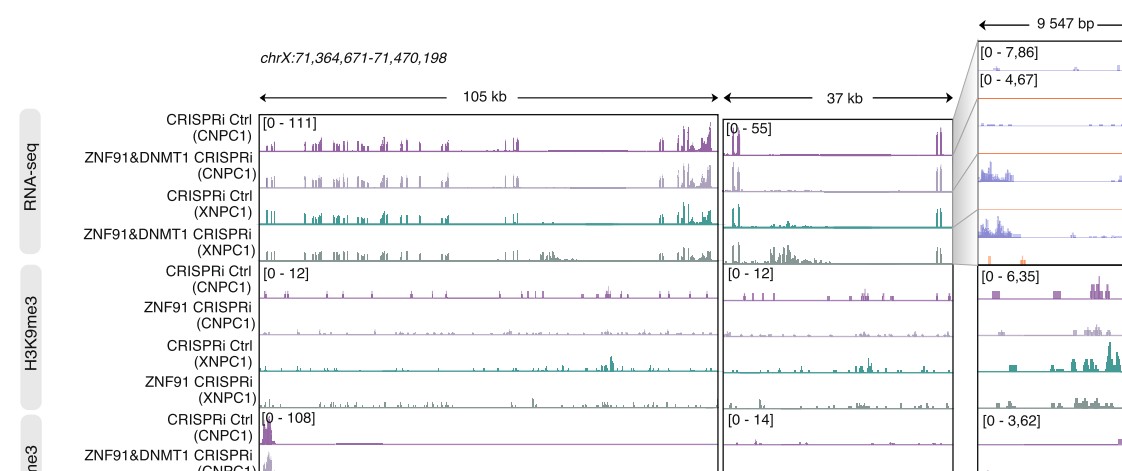

**b** *TAF1* landscape

**Extended Data Fig. 1 | The XDP-NPCs display NPC morphology and expression of NPC markers. a**, Brightfield images of XDP-NPCs and Ctrl-NPCs (left). Immunostainings (right) of Sox2 (green) and Nestin (red) in XDP-NPCs and Ctrl-NPCs. **b**, Genome browser tracks showing RNA-seq, H3K9me3 and H3K4me3 in the *TAF1* gene in CRISPRi-Ctrl, ZNF91-CRISPRi, and ZNF91&DNMT1-CRISPRi conditions in CNPC1 and XNPC1. The reads were mapped to the custom genome, where the XDP-SVA is inserted (See Methods). The XDP-SVA is depicted in red.

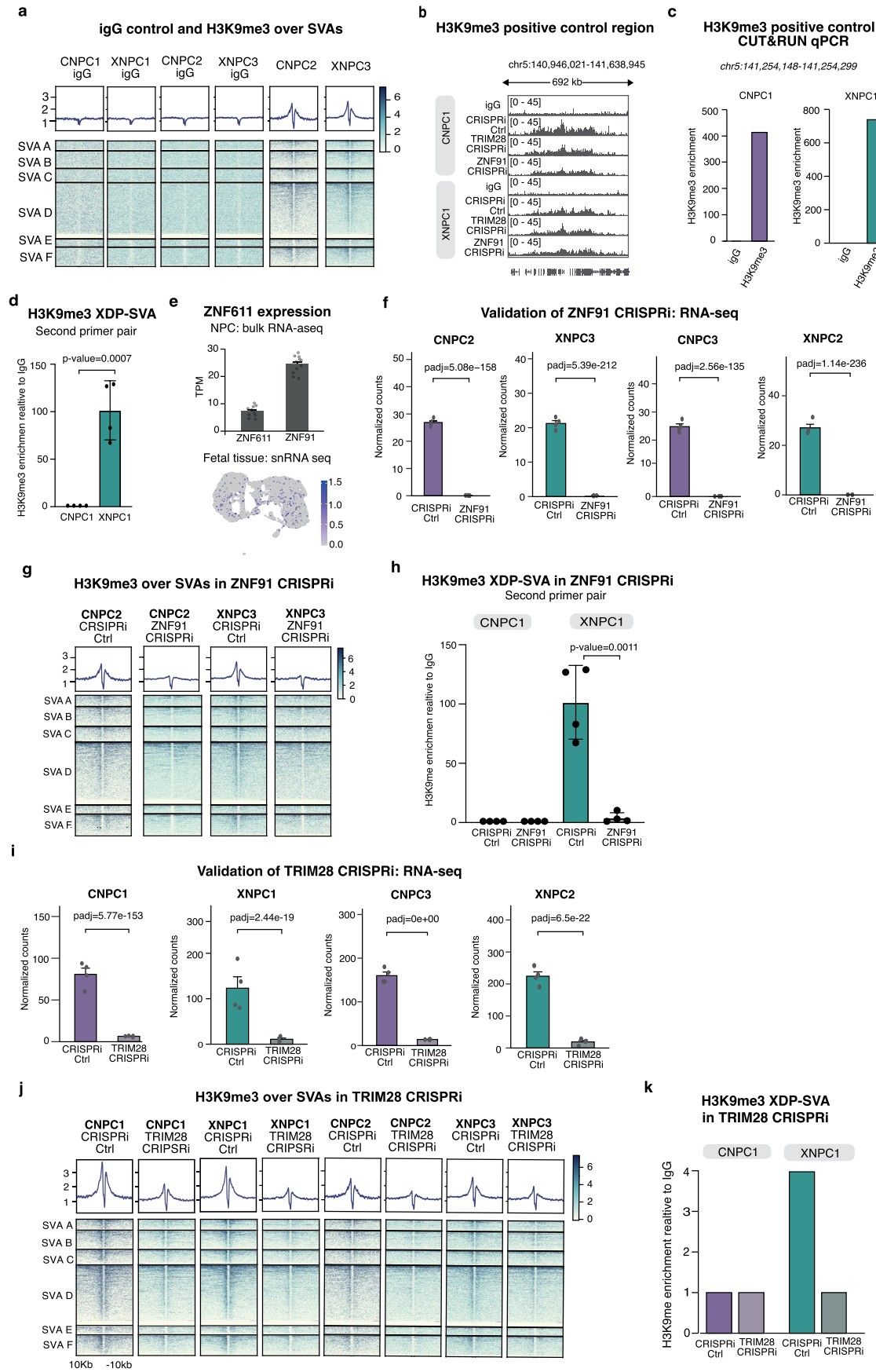

**Extended Data Fig. 2 | See next page for caption.**

**Extended Data Fig. 2 | ZNF91 and TRIM28 orchestrate H3K9me3 deposition over SVAs in NPCs. a**, Heatmap showing igG and H3K9me3 enrichment in Ctrl-NPCs and XDP-NPCs. The genomic regions spanning ±10 kbp from the peak center are displayed. **b**, Genome browser tracks showing H3K9me3 signal over a region known to be covered by H3K9me3 as a positive control. Tracks are shown for Ctrl-NPC and XDP-NPC for H3K9me3 and igG in control, TRIM28, and ZNF91-CRISPRi. **c**, Barplots showing igG and H3K9me3 coverage over a positive-control region using CUT&RUN qPCR in Ctrl-NPC and XDP-NPC (n = 1). **d**, Barplots showing the enrichment of H3K9me3 over the XDP-SVA in XDP-NPCs (n = 4) and the lack of enrichment in Ctrl-NPCs (n = 4, two tailed t-test). A second set of primer pairs were used for this analysis. Bars show H3K9me3 enrichment in each replicate, error bars represent standard deviation. **e**, Barplots comparing *ZNF611* and *ZNF91* expression levels in bulk RNA-seq (n = 12) (top). Bars show the normalized mean expression of the group. UMAP representing *ZNF611* expression (bottom) in different cell types in the fetal brain (n = 3) **f**, Barplots showing normalized mean expression of *ZNF91* (RNA-seq) in CRISPRi-Ctrl and ZNF91-CRISPRi (n = 4). Padj (BH corrected) as calculated by DESeq2 (Wald test, two-sided). Error bars show the standard error of the mean. **g**, Heatmaps showing H3K9me3 signal around SVAs in CRISPRi-Ctrl and ZNF91-CRISPRi (n = 2). **h**, Barplots showing the effect of ZNF91-CRISPRi on H3K9me3 over the XDP SVA in XDP-NPC and Ctrl-NPC (n = 4 in each group, two tailed t-test). A second set of primer pairs were used for this analysis. Bars show H3K9me3 enrichment in each replicate, error bars represent standard deviation. **i**, Barplots showing normalized mean expression of TRIM28 (RNA-seq) in CRISPRi-Ctrl and TRIM28-CRISPRi in Ctrl-NPC and XDP-NPC (n = 4) Padj (BH corrected) as calculated by DESeq2 (Wald test, two-sided). Error bars show the standard error of the mean. **j**, Heatmaps showing H3K9me3 signal around SVAs in CRISPRi-Ctrl and TRIM28-CRISPRi (n = 4). When padj = 0e + 00, it is lower than the computational limit. **k**, Barplots showing the H3K9me3 status of the XDP SVA in CRISPRi-Ctrl and TRIM28-CRISPRi (n = 1).

**a**

**Methylation coverage over the *TAF1* locus**

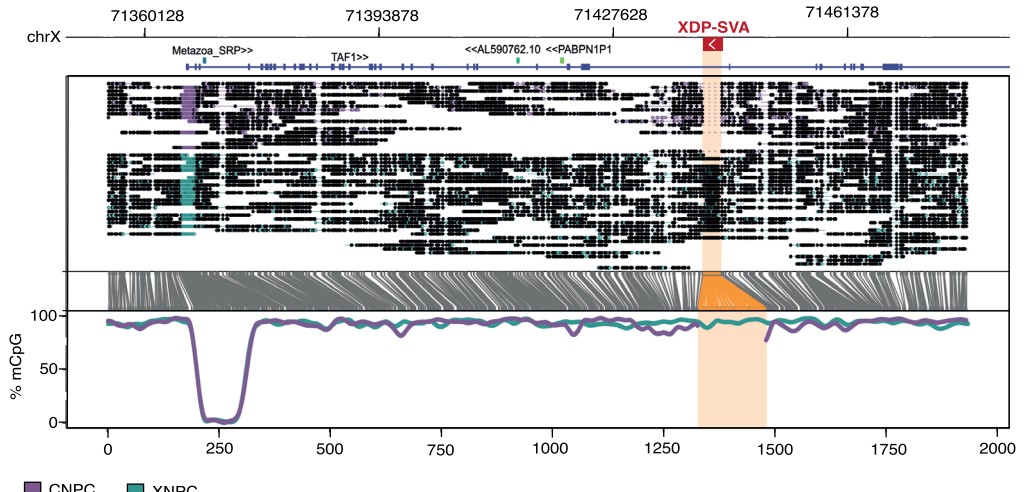

■ CNPC   ■ XNPC

**b**

**5mC immunostaining of DNMT1 KO XNPC1**

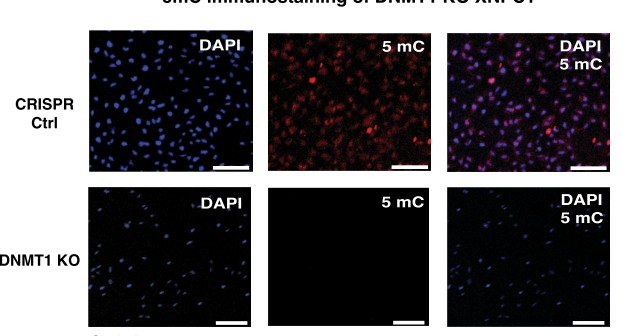

Scale bar = 100um

**c**

**H3K9me3 mark in DNMT1 KO**

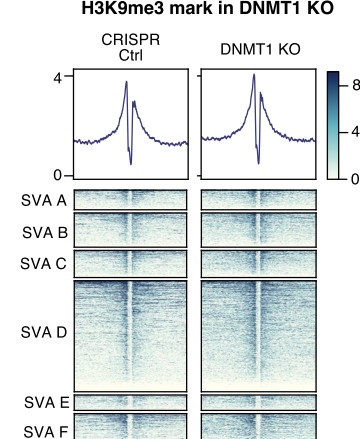

**d**

**H3K4me3 over SVAs in DNMT1&ZNF91 CRISPRi**

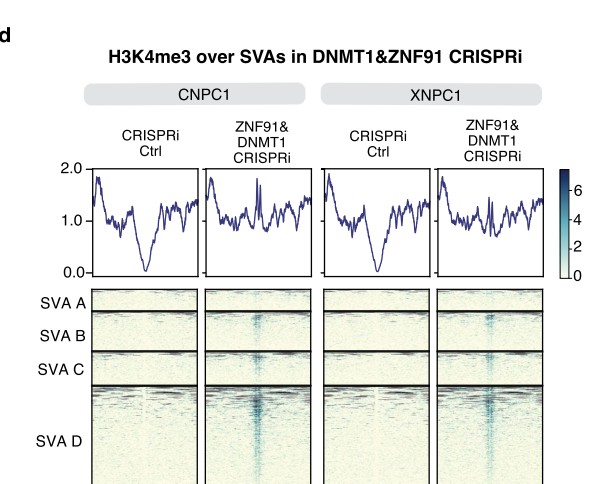

**e**

**H3K4me3 XDP-SVA in ZNF91&DNMT1 CRISPRi**

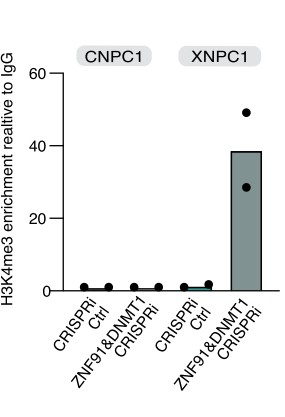

**Extended Data Fig. 3 | See next page for caption.**

**Extended Data Fig. 3 | DNMT1 does not regulate H3K9me3. a**, Methylation coverage over the *TAF1* locus in healthy and XDP-NPCs. **b**, Fluorescent 5mC immunostaining shows successful DNMT1-KO in XNPC1 10 days post transduction. Blue = Dapi, red = 5mC. **c**, Heatmap showing H3K9me3 around SVAs in a genome-wide scale in Ctrl and DNMT1-KO NPCs. **d**, Heatmap showing H3K4me3 enrichment in control and ZNF91&DNMT1-CRISPRi in NPCs. Genomic regions spanning ±10 kbp upstream and downstream of the element are shown. **e**, Bar graphs showing the effect of ZNF91&DNMT1-CRISPRi on H3K4me3 over the XDP SVA in XDP-NPC and Ctrl-NPCs (n = 2 in each group).

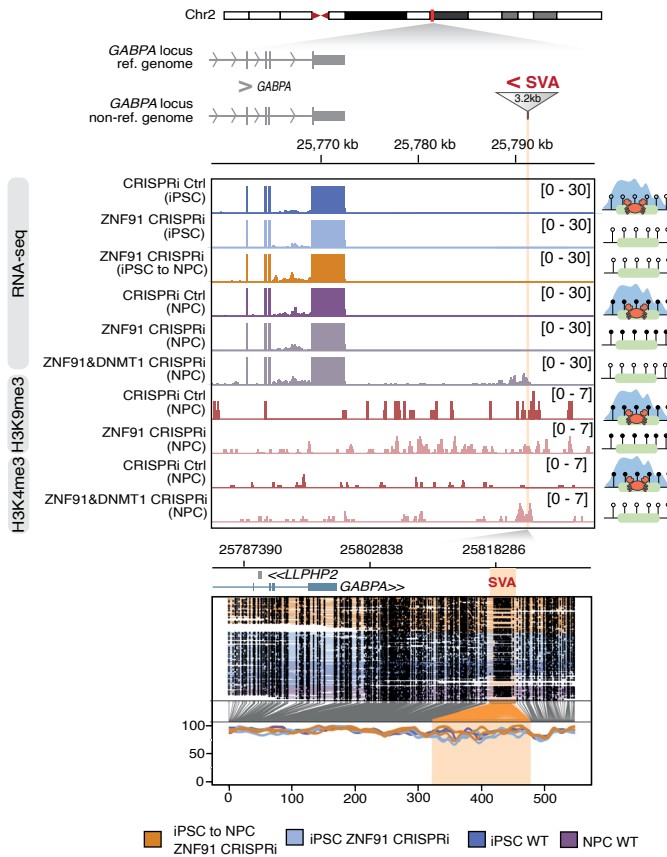

**Extended Data Fig. 4 | ZNF91 independent SVA regulation. a**, Polymorphic SVA loci near *GABPA*. Genome browser tracks (top) showing gene expression, H3K9me3 and H3K4me3. ONT reads (bottom) showing SVA DNA methylation.

# Reporting Summary

## Statistics

For all statistical analyses, confirm that the following items are present in the figure legend, table legend, main text, or Methods section.

| n/a | Confirmed | |
|---|---|---|
| ☐ | ☒ | The exact sample size (*n*) for each experimental group/condition, given as a discrete number and unit of measurement |
| ☒ | ☐ | A statement on whether measurements were taken from distinct samples or whether the same sample was measured repeatedly |
| ☐ | ☒ | The statistical test(s) used AND whether they are one- or two-sided<br>*Only common tests should be described solely by name; describe more complex techniques in the Methods section.* |
| ☐ | ☒ | A description of all covariates tested |
| ☒ | ☐ | A description of any assumptions or corrections, such as tests of normality and adjustment for multiple comparisons |
| ☐ | ☒ | A full description of the statistical parameters including central tendency (e.g. means) or other basic estimates (e.g. regression coefficient) AND variation (e.g. standard deviation) or associated estimates of uncertainty (e.g. confidence intervals) |
| ☐ | ☒ | For null hypothesis testing, the test statistic (e.g. *F*, *t*, *r*) with confidence intervals, effect sizes, degrees of freedom and *P* value noted<br>*Give P values as exact values whenever suitable.* |
| ☒ | ☐ | For Bayesian analysis, information on the choice of priors and Markov chain Monte Carlo settings |
| ☒ | ☐ | For hierarchical and complex designs, identification of the appropriate level for tests and full reporting of outcomes |
| ☒ | ☐ | Estimates of effect sizes (e.g. Cohen's *d*, Pearson's *r*), indicating how they were calculated |

*Our web collection on statistics for biologists contains articles on many of the points above.*

## Software and code

Policy information about availability of computer code

| Data collection | No softwares were used for data collection. |
|---|---|
| Data analysis | All original code has been deposited at GitHub and is publicly available at:<br>https://github.com/raquelgarza/XDP_Horvath_2023<br>The softwares used and mentioned in the manuscript: FeatureCounts (Subread package v1.6.3; hg38 Gencode v38); STAR aligner v2.7.8a ; deeptools v2.5.4; plotHeatmap (v3.5.1); bowtie2 version 2.3.4.2; nanopolish index (v0.13.3) ; EMBOSS cons (v6.6.0.0) ; bedtools getfasta (v2.30.0) ; clustalw2 (v2.1); minimap2 (v2.24) ; TLDR (v1.2.2); nanopolish call-methylation (v0.13.3); methylartist db-nanopolish (v1.2.2); |

For manuscripts utilizing custom algorithms or software that are central to the research but not yet described in published literature, software must be made available to editors and reviewers. We strongly encourage code deposition in a community repository (e.g. GitHub). See the Nature Portfolio guidelines for submitting code & software for further information.

## Data

Policy information about [availability of data](availability of data)

All manuscripts must include a [data availability statement](data availability statement). This statement should provide the following information, where applicable:

- Accession codes, unique identifiers, or web links for publicly available datasets
- A description of any restrictions on data availability
- For clinical datasets or third party data, please ensure that the statement adheres to our [policy](policy)

All processed sequencing data has been deposited and is available at GSE245093 (https://www.ncbi.nlm.nih.gov/geo/query/acc.cgi?acc=GSE245093).
This paper includes analyses of existing, publicly available data. The accession numbers for these datasets are:
GSE224747: 3' single nuclei RNAseq, bulk RNAseq (https://www.ncbi.nlm.nih.gov/geo/query/acc.cgi?acc=GSE224747), and H3K9me3 CUT&RUN of human fetal forebrain tissue. GSE242143: H3K9me3 CUT&RUN from the DNMT1-KO NPCs (https://www.ncbi.nlm.nih.gov/geo/query/acc.cgi?acc=GSE242143)

## Research involving human participants, their data, or biological material

Policy information about studies with [human participants or human data](human participants or human data). See also policy information about [sex, gender (identity/presentation), and sexual orientation](sex, gender (identity/presentation), and sexual orientation) and [race, ethnicity and racism](race, ethnicity and racism).

| | |
|---|---|
| Reporting on sex and gender | The findings of this study apply for male individuals, since the disease we study (X-Linked Dystonia- Parkinsonism) is and X-linked disorder mostly affecting males. The information about the sex of the participants has been collected from WiCell, where the cell lines in use were ordered from. |
| Reporting on race, ethnicity, or other socially relevant groupings | All the individuals used in this study were from Pilipino ancestries since XDP is a disease endemic to those groups. |
| Population characteristics | The age of the XDP patients at collection of the samples in this study varied between 35-72 years, since XDP is an adult-onset disease. The controls used in this study, whenever possible, were the unaffected siblings of the patients and their age at collection was 18-42 years. |
| Recruitment | The participants for this study were recruited by the Collaborative Center for X-Linked Dystonia- Parkinsonism. |
| Ethics oversight | *Identify the organization(s) that approved the study protocol.* |

Note that full information on the approval of the study protocol must also be provided in the manuscript.

# Field-specific reporting

Please select the one below that is the best fit for your research. If you are not sure, read the appropriate sections before making your selection.

☒ Life sciences　　☐ Behavioural & social sciences　　☐ Ecological, evolutionary & environmental sciences

For a reference copy of the document with all sections, see [nature.com/documents/nr-reporting-summary-flat.pdf](nature.com/documents/nr-reporting-summary-flat.pdf)

# Life sciences study design

All studies must disclose on these points even when the disclosure is negative.

| | |
|---|---|
| Sample size | Since there is a limited access to material from XDP patients, no sample size calculations were performed in this study. We used the number of samples that we could access, paying attention to have at least 3 individuals for both control and XDP group. Preliminary experiments using these samples showed that by using 3 XDP and 3 healthy individuals clear conclusions can be reached regarding the results. |
| Data exclusions | No data was excluded from the analysis. |
| Replication | All the generated data has been replicated in at least 3 replicates, in most of the cases 4 replicates. The findings were replicable. |
| Randomization | To randomize the experimental groups, we used code names for the cell lines, without indication if they belonged to the XDP or control group. |
| Blinding | The investigators were blinded to group allocation, since code names were used as sample identifiers without containing the information regarding the state of the sample (XDP or control). |

# Reporting for specific materials, systems and methods

We require information from authors about some types of materials, experimental systems and methods used in many studies. Here, indicate whether each material, system or method listed is relevant to your study. If you are not sure if a list item applies to your research, read the appropriate section before selecting a response.

## Materials & experimental systems

| n/a | Involved in the study |
|---|---|
| ☐ | ☒ Antibodies |
| ☐ | ☒ Eukaryotic cell lines |
| ☒ | ☐ Palaeontology and archaeology |
| ☒ | ☐ Animals and other organisms |
| ☒ | ☐ Clinical data |
| ☒ | ☐ Dual use research of concern |
| ☒ | ☐ Plants |

## Methods

| n/a | Involved in the study |
|---|---|
| ☒ | ☐ ChIP-seq |
| ☒ | ☐ Flow cytometry |
| ☒ | ☐ MRI-based neuroimaging |

## Antibodies

| Antibodies used | 5mC, Active Motif, cat.no. 39649, lot 02617020, used 1:250; SOX2, R&D Systems, AF2018, 1:100; Nestin, Abcam, AB176571, 1:100; donkey anti-rabbit Alexa fluor 647, Jackson Lab, 711-605-152, lot: 167518, 1:200; donkey anti-goat cy3, Jackson Lab, 705-165-003, lot: 156134, 1:200; rabbit anti H3K9me3, Abcam ab8898, RRID:AB_306848, 1:50; rabbit anti H3K4me3, Active Motif cat#39159, RRID:AB_2555751 , 1:50 |
|---|---|
| Validation | 5mC<br>Validated Applications by the manufacturer: MeDIP: 1 μg per IP, IHC (FFPE): 1:1000, ELISA: 1:10,000 dilution. The following applications have been published using this antibody. Unless noted above, Active Motif may not have validated the antibody for use in these applications: MeDIP, MeDIP-Seq, ICC/IF , Flow Cytometry, IHC(FFPE), DB<br>https://www.activemotif.com/catalog/details/39649/5-methylcytosine-5-mc-antibody-mab-clone-33d3<br><br>SOX2<br>The antibody has been validated in 201 citations based on the manufacturer's website. such as Campbell et al. Int J Mol Sci. 2023 Feb 9;24(4):3477. doi: 10.3390/ijms24043477. Among others, SOX2 was also detected in immersion fixed ADLF1 (top panel) and FAB2 (bottom panel) induced pluripotent stem cell lines using Goat Anti-Human/Mouse/Rat SOX2 Antigen Affinity-purified Polyclonal Antibody (Catalog # AF2018) at 10 μg/mL for 3 hours at room temperature.<br>https://www.rndsystems.com/products/human-mouse-rat-sox2-antibody_af2018?gad_source=1&gclid=CjwKCAjwh4-wBhB3EiwAeJsppILQ5DJh7iJa7jFq2WSC7WPUpnPYPiCjX5opEj1zWiG0YYL7AOw8_hoCPhUQAvD_BwE&gclsrc=aw.ds<br><br>Nestin<br>Based on the manufacturer's website the antibody was validated in IHC-P, ICC/IF, Flow Cyt (Intra) and tested in Human samples. Cited in 5 publications such as https://pubmed.ncbi.nlm.nih.gov/27016413/<br>https://www.abcam.com/en-mx/products/primary-antibodies/nestin-antibody-epr13012-ab176571#application=ihc-p<br><br>H3K9me3<br>Rabbit Polyclonal H3 tri methyl K9 antibody. Validated in IHC-P, ICC/IF, ChIP, WB and tested in Human, Mouse, Cow samples. Cited in 1345 publications. e.g. https://pubmed.ncbi.nlm.nih.gov/23001792/<br>https://www.abcam.com/en-se/products/primary-antibodies/histone-h3-tri-methyl-k9-antibody-chip-grade-ab8898#application=chip<br><br>H3K4me3<br>Applications Validated by Active Motif: ChIP: 3 - 5 μl per ChIP; ChIP-Seq: 3 μl each; ICC/IF: 1:500 - 1:1,000 dilution; WB: 1:500 - 1:2,000 dilution; CUT&Tag: 1 μl per 50 μl reaction*; CUT&RUN: 1 μl per 50 μl reaction<br><br>*This antibody has been validated for CUT&Tag using Active Motif's CUT&Tag-IT™ Assay Kit, Catalog No. 53160.<br>modENCODE validation: this antibody was validated for ChIP-Seq in this study (see reference).<br>NGS-QC® certification: this antibody has been processed by the NGS-QC® generator.<br><br>The following applications have been published using this antibody. Unless noted above, Active Motif may not have validated the antibody for use in these applications: ChIP-qPCR, ChIP-chip, ChIP-Seq; Native ChIP; CUT&RUN, CUT&Tag; WB; IF; FC; IHC(P); ELISA Proximity Ligation Assay (PLA)<br><br>https://www.activemotif.com/catalog/details/39159 |

## Eukaryotic cell lines

Policy information about cell lines and Sex and Gender in Research

| Cell line source(s) | All cell lines used in this study were derived from human participants and were obtained from WiCell. All samples were from male individuals. |
|---|---|

| Authentication | The authentication of the XDP and healthy cell lines was done by RNA-seq analysis by identifying the XDP specific TAF1 molecular phenotype. |
|---|---|
| Mycoplasma contamination | The cell lines were regularly tested for mycoplasma contamination and the test results were always negative. |
| Commonly misidentified lines (See ICLAC register) | There were no commonly misidentified lines in this study. |

## Plants

| Seed stocks | *Report on the source of all seed stocks or other plant material used. If applicable, state the seed stock centre and catalogue number. If plant specimens were collected from the field, describe the collection location, date and sampling procedures.* |
|---|---|
| Novel plant genotypes | *Describe the methods by which all novel plant genotypes were produced. This includes those generated by transgenic approaches, gene editing, chemical/radiation-based mutagenesis and hybridization. For transgenic lines, describe the transformation method, the number of independent lines analyzed and the generation upon which experiments were performed. For gene-edited lines, describe the editor used, the endogenous sequence targeted for editing, the targeting guide RNA sequence (if applicable) and how the editor was applied.* |
| Authentication | *Describe any authentication procedures for each seed stock used or novel genotype generated. Describe any experiments used to assess the effect of a mutation and, where applicable, how potential secondary effects (e.g. second site T-DNA insertions, mosiacism, off-target gene editing) were examined.* |

