## [Peer Review File · Nature Structural & Molecular Biology]

Peer Review Information

Manuscript Title: Mini-heterochromatin domains constrain the cis-regulatory impact of SVA transposons in human brain development and disease

Corresponding author name(s): Johan Jakobsson

Reviewer Comments & Decisions:

Decision Letter, initial version:
--

Message: 5th Jan 2024

Dear Professor Jakobsson,

Thank you again for submitting your manuscript "Mini-heterochromatin domains constrain the cis-regulatory impact of SVA transposons in human brain development and disease". I apologise for the delay in responding, which resulted from the difficulty in timely obtaining suitable referee reports due to the winter holidays. Nevertheless, we now have comments (below) from the 3 reviewers who evaluated your paper. In light of these reports, we remain interested in your study and would like to see your response to the comments of the referees, in the form of a revised manuscript.

You will see that though all reviewers are appreciative of the findings, they also raise several concerns that need to be addressed in a revised manuscript. More specifically, the experts request clarifications, additional controls, more extensive contextualisation and discussion within existing literature, further experiments and analyses at relevant parts of the manuscript. Both reviewers #1 and #3 note the lack of relevant data and discussion of the hexanucleotide repeat within the SVA element that regulates onset of XDP, and they request the addition of relevant analyses for other polymorphic SVA insertions (Reviewer #3 point 3) and for global SVA detection and global interplay with TRIM28/ZNF91 (last point of Reviewer #1). Finally, the same experts, though understanding the chosen focus of the paper, express the desire for more mechanistic details. Though this is not a strict requirement for the success of the story, we editorially agree with the experts about the fact that further mechanistic insight would elevate the value of the manuscript and would encourage you to try to provide such insight with relevant experiments and analyses, in accordance to the guidance/questions posed by the referees.

Please be sure to address/respond to all concerns of the referees in full in a point-by-point response and highlight all changes in the revised manuscript text file. If you have

comments that are intended for editors only, please include those in a separate cover letter.

We expect to see your revised manuscript within 3 months. If you cannot send it within this time, please contact us to discuss an extension; we would still consider your revision, provided that no similar work has been accepted for publication at NSMB or published elsewhere.

Reporting Summary:

When submitting the revised version of your manuscript, please pay close attention to our [href="https://www.nature.com/nature-portfolio/editorial-policies/image-integrity">Digital Image Integrity Guidelines](https://www.nature.com/nature-portfolio/editorial-policies/image-integrity). and to the following points below:

Data availability: this journal strongly supports public availability of data. All data used in accepted papers should be available via a public data repository, or alternatively, as Supplementary Information. If data can only be shared on request, please explain why in your Data Availability Statement, and also in the correspondence with your editor. Please note that for some data types, deposition in a public repository is mandatory - more information on our data deposition policies and available repositories can be found below: <https://www.nature.com/nature-research/editorial-policies/reporting-standards#availability-of-data>

Nature Structural & Molecular Biology is committed to improving transparency in authorship. As part of our efforts in this direction, we are now requesting that all authors identified as 'corresponding author' on published papers create and link their Open Researcher and Contributor Identifier (ORCID) with their account on the Manuscript Tracking System (MTS), prior to acceptance. This applies to primary research papers only. ORCID helps the scientific community achieve unambiguous attribution of all scholarly contributions. You can create and link your ORCID from the home page of the MTS by clicking on 'Modify my Springer Nature account'. For more information please visit please visit www.springernature.com/orcid.

[Redacted]

Sincerely,

Dimitris Typas
Associate Editor
Nature Structural & Molecular Biology
ORCID: 0000-0002-8737-1319

Reviewers' Comments:

Reviewer #1:

Remarks to the Author:

Horvath et al. investigated SVA retrotransposon activity and how global DNA methylation and/or specific H3K9me3 can influence SVA expression. They find that ZNF91 established H3K9me3 in somatic neural progenitor cells (NPCs) and affects downstream SVA expression. They also find that DNA CpG methylation is not affected by ZNF91 in somatic NPCs. However, CpG methylation is affected by ZNF91 in iPSCs while differentiating into NPCs. The authors show that this is a global process that affects SVAs in a cis-regulatory manner. Additionally, the authors suggest that local heterochromatin aggravates the XDP phenotype. The manuscript was an intriguing read. There are a few concerns from the manuscript where the authors' conclusions and interpretations might be questionable. Overall, the disease mechanisms of TAF1 is not specifically further unraveled in the study, the focus is rather on a general mechanism of how most SVAs are repressed.

The author's are understandably interested in the role of TAF1 expression in the context of SVA retrotransposon exploration and TAF1 expression written as a solid XDP disease mechanism. Though there are strong suggestions, TAF1 gene expression is not the only mechanism that has been proposed to affect XDP. Importantly, there is a hexanucleotide repeat expansion that influences disease onset that may form RNA foci, there are also other differentially expressed transcripts. I encourage the authors to discuss this in the introduction as to not oversimplify the picture. These other mechanisms are just not explored in their manuscript but does not mean they are not a predominant disease target/mechanism.

The authors elegantly show their Crispr knockdown functions well. It would also be helpful to show the exact location of where the H3k9me3 is in boundaries are around the SVA (a genomic pictorial zooming into Fig 1d would be sufficient).

The authors write ZNF91 and ZNF611 as well as several other KZFPs bind to SVAs. However they focus on ZNF91. Can the authors comment on why not investigate the others? Can the authors show ZNF611 expression in Figure 2g as well to show that there is difference (or not)? It is already known that ZNF91 deletion leads to the activation of SVA retrotransposons (Haring et al 2021 Genome Research) and that there is a decrease in H3K9me3 when this happens. Thus, this part is not entirely novel.

The CpG methylation around the SVA has also already been shown (Luth et al Genes 2022) and specifically in the brain however tissue types can have some variation in methylation. Can the authors discuss these findings in the context of their own given the fact that there is no post-mortem investigated?

The authors write that while in somatic cells only H3K9me3 depends on ZNF91, DNA methylation is propagated through other mechanisms, indicated that it can happen in early development. I might be a bit more cautious in this conclusion/interpretation as there is differentiation process in vitro that happens and the process can be quite artificial. Another confirmatory method could be to check mouse embryonic development and look at stages to really show this DNA methylation happens in early development.

Can the authors show the DNMT1 CRISPRi in iPSC to NPC (as part of Figure 4e)?

There needs to be more XDP-related details and investigation. The authors did not comment on the TSS of TAF1 in XDP and how far it is from the SVA? Where is the heterochromatin domain in TAF1? Can the authors remove the specific heterochromatin domain at the XDP SVA to check for SVA expression and TAF1 expression? At the moment

the focus is global expression as ZNF91 affects other SVAs and further experiments are required to confirm this local effect.

Lastly the authors give two examples in Fig. 6. It would be important for the reader to understand globally how many SVAs they can detect? Are there some SVAs where methylation and H3K9me3 is not influenced by TRIM28 or ZNF91? Is the methylation of the surrounding DNA influenced by the SVA insertions? Perhaps a summary of all SVAs in Fig. 6 would give a better overview and support their conclusions in the abstract.

Reviewer #2:

Remarks to the Author:

In this manuscript from Horvath and colleagues, the authors establish neural progenitor cell models from iPSCs derived from three patients with XDP-SVA and controls (unaffected sons of two of the carriers). The authors demonstrate that these cultures have characteristic and defining features of XDP, confirming the presence of the SVA insertion in TAF1, with reduced exon 38 expression and retention of intron 32. The authors provide evidence that SVAs are marked genome-wide (and at the XDP-SVA element) by H3K9me3, and that this mark is dependent on ZNF91/Trim28, which were depleted using CRISPRi, consistent with previous studies implicating these factors in SVA binding and repression. Using nanopore sequencing, the authors confirm that SVAs are heavily methylated in NPCs, including the polymorphic SVA in XDP, and that this methylation is independent of ZNF91 in NPCs. The authors then go on to show that the establishment of methylation of SVAs is likely at least partially dependent on ZNF91, as depletion of ZNF91 in iPSCs prior to differentiation into NPCs leads to hypomethylation of SVAs. The authors then perform experiments to perturb DNA methylation via removal of Dnmt1, and reveal important separable functions for DNA methylation and ZNF91/TRIM28 dependent H3K9me3 for SVA silencing. Finally the authors show that the loss of ZNF91 and Dnmt1 leads to misregulation of genes nearby SVA elements, including at the TAF1 locus in patients with XDP, leading support to the hypothesis that one important function of these SVA restriction pathways is to limit the cis-regulatory effects of TE insertions.

Although it has already been shown that ZNF91 and DNA methylation play a role in SVA recognition and silencing, this study adds to the growing body of work KRAB-ZNF/KAP1 and DNA methylation play parallel functions to limit the regulatory impact of TEs, and that this has important consequences in human diseases caused by SVA insertion, including XDP. The study was nicely designed, uses elegant genetics/CRISPR and neural stem cell cultures from patients derived iPSCs. It will be of interest to a broad audience interested in epigenetics, human neural development and disease, and TE regulation.

Issues that need to be addressed.

In figures 2D and 2K, the authors should use additional primer pairs at regions flanking (both upstream and downstream) the XDP-SVA to confirm that H3K9me3 is enriched relative to adjacent regions and that there isn't simply a difference in overall enrichment across the samples (which have to be treated separately because they are two different cell lines). The authors need to also provide error bars and statistics to show that these differences are statistically significant.

Reviewer #3:

Remarks to the Author:

This study aims to dissect the mechanisms that control the regulatory impact of SVA retrotransposons in the human genome, focussing on the interesting case of an intronic SVA insertion in the TAF1 gene that causes X-linked dystonia-parkinsonism (XDP). They conduct a series of experiments using ips cells and derived neural progenitor cells (NPCs) to show that, in most cases, both H3K9me3 (controlled by ZNF91) and DNA methylation are required to repress general SVA expression and their regulatory effects on nearby genes. In the case of the SVA in the intron of TAF1, they find that loss of these repressive marks causes a higher degree of partial intron retention and lower level of expression of a downstream exon, indicating a more pronounced molecular defect. They then identify young, insertionally polymorphic SVA elements in their cell lines and examine transcriptional effects when H3K9me3 and DNA methylation are knocked down. The paper was well written with appropriate references and the experiments were generally clear and well controlled. The quality of the data is high and the approaches and statistics valid.

Comments:

1. The fact that most SVAs are heavily methylated, bind ZNF91 and are marked with H3K9me3 has been reported before, so the main novelty or originality of this study is demonstrating the effect on expression of TAF1 when the repressive marks on the SVA are removed, with the results shown in figures 5e and 5d. This is an interesting finding, and it may help explain the late age of onset of XDP since, as the authors mention, it is known that DNA methylation levels decrease in the aging brain. However, it was somewhat disappointing that no further insight was gained into the actual mechanism by which the SVA causes the transcriptional defect of TAF1 in the first place.
2. In figure 1f, please show the location of the SVA element in the genome browser tracks and indicate its transcriptional orientation with respect to the TAF1 gene. A quick perusal of the 2018 Cell paper describing this mutation found that the spurious intronic TAF1 transcription terminated 5' to the SVA. It should be made clear if this present study also found the same thing. Is this SVA full length with an intact promoter? Is there any evidence of transcription initiating or terminating in that SVA when H3K9me3 and DNA methylation are removed? Perhaps I missed it, but I could not find mention of this in the manuscript.
3. In the latter section of the paper, the authors aim to examine the epigenetic marks on other polymorphic (recent) SVA insertions in two of their cell lines. They state (on page 21): "We identified 22 high confidence polymorphic insertions of the SVA-E and -F subfamilies (average 2.5 kbp in length, range 1.23.8 kbp), of which 14 were shared between the two genomes (Fig.6b). Notably, several of these polymorphic SVAs, which represent recent TE insertions into the germline of these two individuals, displayed clear hallmarks of local heterochromatinization, including the presence of DNA methylation and H3K9me3." Then they show two examples in Figures 6c and d. Are these just the most exceptional cases? For how many of the 22 insertions were they able to identify marks of heterochromatinization? The authors should give some indication of the generality of their findings.

4. In Figure 6c, the RNA-seq tracks are confusing until one realizes that the EMC4 gene is oriented opposite to SLC12A6. Please make this clear in the figure. Similarly, in Figure 6d, the top part of the figure seems to show ATP5PF and GABPA oriented in the same direction but in fact these two genes are oriented in opposite directions and appear to be using a bidirectional promoter (according to Genome Browsers). Once this is clarified, the stranded RNA-seq tracks will be easier to understand.

5. For these same two examples, the RNA-seq seems to indicate that the SVAs are being transcribed when the repressive epigenetic marks are removed but the authors show no other data to support this. Are these SVAs actually close to full length with structurally intact promoters? Were these the only two examples where evidence of polymorphic SVA transcription upon release from epigenetic repression was observed?

6. It has been reported in the literature that length of a (CCCTCT)_n repeat within the SVA element is an age-at-onset modifier in XDP. Perhaps it would be worth mentioning if the cell line models used in this study harbor repeat sizes typically found in patients.

Author Rebuttal to Initial comments

Point-by-point response, Horvath et al. NSMB-A48559A

Reviewers' Comments:

Reviewer #1:

Remarks to the Author:

Horvath et al. investigated SVA retrotransposon activity and how global DNA methylation and/or specific H3K9me3 can influence SVA expression. They find that ZNF91 established H3K9me3 in somatic neural progenitor cells (NPCs) and affects downstream SVA expression. They also find that DNA CpG methylation is not affected by ZNF91 in somatic NPCs. However, CpG methylation is affected by ZNF91 in iPSCs while differentiating into NPCs. The authors show that this is a global process that affects SVAs in a cis-regulatory manner. Additionally, the authors suggest that local heterochromatin aggravates the XDP phenotype. The manuscript was an intriguing read. There are a few concerns from the manuscript where the authors' conclusions and interpretations might be questionable. Overall, the disease mechanisms of TAF1 is not specifically further unraveled in the study, the focus is rather on a general mechanism of how most SVAs are repressed.

We would like to thank the reviewer for the overall positive comments on our manuscript. We have now generated a new version of the manuscript with revisions in line with the suggestions from the reviewers. We genuinely feel that

the manuscript is greatly improved. Changes in the manuscript are outlined in the point-by-point response below and can be found in red font in the new version of the manuscript.

The author's are understandably interested in the role of TAF1 expression in the context of SVA retrotransposon exploration and TAF1 expression written as a solid XDP disease mechanism. Though there are strong suggestions, TAF1 gene expression is not the only mechanism that has been proposed to affect XDP. Importantly, there is a hexanucleotide repeat expansion that influences disease onset that may form RNA foci, there are also other differentially expressed transcripts. I encourage the authors to discuss this in the introduction as to not oversimplify the picture. These other mechanisms are just not explored in their manuscript but does not mean they are not a predominant disease target/mechanism.

We agree with the reviewer that there are several potential mechanisms by which the SVA insertion could influence XDP pathology. In line with the suggestion from the reviewer we have added a new sentence to the introduction on page 4 in which we discuss this issue. This sentence is also inserted below.

'It is unclear how the SVA insertion in TAF1 leads to XDP pathology. The presence of the SVA has been linked to alternative splicing and decreased transcripts levels of TAF1, which may be a direct consequence of intron retention of the 32nd intron (PMID: 29474918, PMID: 31116117, PMID:26769779). In addition, the XDP-SVA contains an unstable hexameric repeat that displays polymorphic variation linked to age of onset, and with the potential to be expanded in somatic tissues. Such expanded hexameric repeats have been speculated to be the source of toxic transcripts or to induce transcriptional interference of TAF1(PMID: 29229810).'

The authors elegantly show their Crispri knockdown functions well. It would also be helpful to show the exact location of where the H3k9me3 is in boundaries are around the SVA (a genomic pictorial zooming into Fig 1d would be sufficient).

Due to mappability issues and the structure of the nearby genome it is very challenging to estimate the presence of different histone marks, such as H3K9me3, over and at the boundaries of the XDP-SVA. We therefore developed a qPCR-based technique in combination with CUT&RUN. This approach is more sensitive than sequencing and allows us to estimate H3K9me3 at the XDP-SVA boundaries. We have added a new sketch that clarifies this approach in Fig 2d. This sketch is also inserted below.

The authors write ZNF91 and ZNF611 as well as several other KZFPs bind to SVAs. However they focus on ZNF91. Can the authors comment on why not investigate the others? Can the authors show ZNF611 expression in Figure 2g as well to show that there is difference (or not)? It is already known that ZNF91 deletion leads to the activation of SVA retrotransposons (Haring et al 2021 Genome Research) and that there is a decrease in H3K9me3 when this happens. Thus, this part is not entirely novel.

We agree with the reviewer that several other KZFPs have been implicated in SVA-binding (PMID: 28273063). Of these KZFPs, ZNF91 and ZNF611 has been validated to play a functional role in SVA repression (PMID: 33722937; PMID: 31006620). Thus, we hypothesised that these two KZFPs are prime candidates to bind the XDP-SVA in neural cells. When we compared the expression levels of ZNF91 and ZNF611 in NPCs and human brain development it became clear that ZNF91 had a higher expression, suggesting that this is the most interesting KZFP in this context. This was then confirmed in our CRISPRi experiments where deletion of ZNF91 resulted in an almost complete loss of H3K9me3 at SVAs. In the new version of the manuscript, we have added the comparison of ZNF91 and ZNF611 expression levels in NPCs and human brain development to the supplementary information (new Extended Data Figure 2e). We have also added a brief comment in the results section on p. 9. The new figure is also inserted below.

Extended Data Fig2. e, Barplots comparing ZNF611 and ZNF91 expression levels in bulk RNA-seq from NPCs (top). UMAP representing ZNF611 expression (bottom) in different cell types in the fetal brain

The CpG methylation around the SVA has also already been shown (Luth et al Genes 2022) and specifically in the brain however tissue types can have some variation in methylation. Can the authors discuss these findings in the context of their own given the fact that there is no post-mortem investigated?

This is a good point. There is emerging data on the methylation status of SVAs in the human brain, including the XDP-SVA, that should be mentioned in the discussion in relation to our findings. We have added a couple of sentences on this topic to the discussion on p.28. These sentences are also inserted below.

‘Our data are limited to cell culture models; the epigenetic status of the XDP-SVA in human brain tissue has not been extensively studied. However, long-read ONT data obtained from post-mortem brain tissue of one individual suggest that SVAs are highly methylated in the adult brain, although the methylation level appears to be slightly lower in polymorphic SVA insertions (PMID: 33186547). In addition, the presence of DNA methylation over the XDP-SVA was investigated in post-mortem brain tissue from a patient (PMID: 35052466). This analysis showed that although the XDP-SVA is methylated in brain tissue, the methylation levels in the brain were reduced compared to blood samples from the same individual. It is worth noting that DNA methylation patterns in the human brain change with age (PMID: 31462381, PMID: 21216877, PMID: 22305529). It will be interesting to investigate whether DNA methylation over the XDP-SVA is stable in the human brain, or

whether it is lost with age. Such phenomena could explain the late-onset phenotype of XDP, where a gradual increase in the loss of TAF1 function ultimately results in cellular dysfunction. Such a scenario would also open new therapeutic possibilities where restoration of DNA methylation on the XDP-SVA could block or reverse the pathology. However, further analysis using larger cohorts of post-mortem XDP tissue is required to investigate this possibility. ‘

The authors write that while in somatic cells only H3K9me3 depends on ZNF91, DNA methylation is propagated through other mechanisms, indicated that it can happen in early development. I might be a bit more cautious in this conclusion/interpretation as there is differentiation process in vitro that happens and the process can be quite artificial. Another confirmatory method could be to check mouse embryonic development and look at stages to really show this DNA methylation happens in early development.

We agree with the reviewer. This is an important point. We have added a statement to the discussion on page 26. This statement is also inserted below.

The idea to investigate this phenomenon in mouse models is an interesting and could be pursued in the future. If mice that encode the human version of TAF1, including the SVA insertion, as well as ZNF91 become available it would be possible to model this interaction in an *in vivo* setting. However, the generation of such mouse models will be both challenging and time consuming and lie well outside the scope of this study.

‘There are limitations to these conclusions. Although the iPSC-NPC model system used in this project recapitulates some aspects of human brain development, it is a simplified model of early brain development. To validate some of the findings in this study we have relied on human fetal tissue. However, such analysis does not provide much mechanistic insight and we cannot exclude the possibility that some of our observations are in vitro artefacts. In addition, the iPSC-NPC system does not allow us to investigate differences in the epigenetic silencing of SVAs over time and in different cell types. It will be interesting to address such questions using more complex model systems, such as cerebral organoids (PMID: 36171373). ‘

Can the authors show the DNMT1 CRISPRi in IPSC to NPC (as part of Figure 4e)?

Surprisingly, and in contrast to findings in mouse, deletion of DNMT1 is not tolerated in human pluripotent stem cells such as ES or iPSCs. Deletion of DNMT1

in human pluripotent stem cells result in rapid cell death. This is described in a beautiful paper from Alexander Meissner and collaborators (PMID: 25822089). Thus, we have not been able to perform the DNMT1 CRISPRi in human iPSCs.

There needs to be more XDP-related details and investigation. The authors did not comment on the TSS of TAF1 in XDP and how far it is from the SVA? Where is the heterochromatin domain in TAF1? Can the authors remove the specific heterochromatin domain at the XDP SVA to check for SVA expression and TAF1 expression? At the moment the focus is global expression as ZNF91 affects other SVAs and further experiments are required to confirm this local effect.

The XDP-SVA is located 74.2 kbp from the *TAF1* TSS. We find no evidence that the H3K9me3 spreads to the TSS. On the contrary, ZNF91-deposited H3K9me3 appears to be very locally distributed at SVAs (see Fig.2b). Using ONT DNA methylation analysis we also find no evidence that DNA methylation spreads from the XDP-SVA to the promoter region of TAF1. We now mention this on p. 6 and p.13 in the result section and the extended TAF1 DNA methylation analysis is inserted into Extended Data Fig3a. It is also inserted below.

We also agree with the reviewer that our strategy to delete ZNF91 and DNMT1 results in a global effect on SVAs. However, since we are using iPSCs derived from the close relatives of the XDP-patients, we can be quite certain when the effect is due to the local effects of the XDP-SVA.

a

Extended data Fig. 3: a, Methylation coverage over the *TAF1* locus in Ctrl-NPCs and XDP-NPCs.

Lastly the authors give two examples in Fig. 6. It would be important for the reader to understand globally how many SVAs they can detect? Are there some SVAs where methylation and H3K9me3 is not influenced by TRIM28 or ZNF91? Is the methylation of the surrounding DNA influenced by the SVA insertions? Perhaps a summary of all SVAs in Fig. 6 would give a better overview and support their conclusions in the abstract.

We agree with the reviewer that this is an important point. We have now collected this information into a new supplementary table. This table includes information on the location and size of the SVA insertion as well as the DNA-methylation level. It also includes information on if the SVA is flanked by H3K9me3 or H3K4me3 (see response#5, reviewer#3 for details on this experiments) and if these marks are dependent on ZNF91. Lastly, it contains information on if we can detect transcripts originating from the SVA upon ZNF91/DNMT1 CRISPRi. All this information is found in the new Supplementary Table 1. We mention this new table in the text on p.23 and 30.

Reviewer #2:

Remarks to the Author:

In this manuscript from Horvath and colleagues, the authors establish neural progenitor cell models from iPSCs derived from three patients with XDP-SVA and controls (unaffected sons of two of the carriers). The authors demonstrate that these cultures have characteristic and defining features of XDP, confirming the presence of the SVA insertion in TAF1, with reduced exon 38 expression and retention of intron 32. The authors provide evidence that SVAs are marked genome-wide (and at the XDP-SVA element) by H3K9me3, and that this mark is dependent on ZNF91/Trim28, which were depleted using CRISPRi, consistent with previous studies implicating these factors in SVA binding and repression. Using nanopore sequencing, the authors confirm that SVAs are heavily methylated in NPCs, including the polymorphic SVA in XDP, and that this methylation is independent of ZNF91 in NPCs. The authors then go on to show that the establishment of methylation of SVAs is likely at least partially dependent on ZNF91, as depletion of ZNF91 in iPSCs prior to differentiation into NPCs leads to hypomethylation of SVAs. The authors then perform experiments to perturb DNA methylation via removal of Dnmt1, and reveal important separable functions for DNA methylation and ZNF91/TRIm28 dependent H3K9me3 for SVA silencing. Finally the authors show that the loss of ZNF91 and Dnmt1 leads to misregulation of genes nearby SVA elements, including at the TAF1 locus in patients with XDP, leading support to the hypothesis that one important function of these SVA restriction pathways is to limit the cis-regulatory effects of TE insertions.

Although it has already been shown that ZNF91 and DNA methylation play a role in SVA recognition and silencing, this study adds to the growing body of work KRAB-ZNF/KAP1 and DNA methylation play parallel functions to limit the regulatory impact of TEs, and that this has important consequences in human diseases caused by SVA insertion, including XDP. The study was nicely designed, uses elegant genetics/CRISPR and neural stem cell cultures from patients derived iPSCs. It will be of interest to a broad audience interested in epigenetics, human neural development and disease, and TE regulation.

We would like to thank the reviewer for the interest in our manuscript and for the positive comments. We have now generated a new version of the manuscript with revisions in line with the suggestions from the reviewers. We genuinely feel that the manuscript is greatly improved. Changes in the manuscript are outlined in the point-by-point response below and can be found in red font in the new version of the manuscript.

Issues that need to be addressed.

In figures 2D and 2K, the authors should use additional primer pairs at regions flanking (both upstream and downstream) the XDP-SVA to confirm that H3K9me3 is enriched relative to adjacent regions and that there isn't simply a difference in overall enrichment across the samples (which have to be treated separately because they are two different

cell lines). The authors need to also provide error bars and statistics to show that these differences are statistically significant.

We have repeated this experiment with more replicates (n=4 for each group). This new analysis confirms that the H3K9me3 enrichment at the XDP-SVA is statistically significant (two tailed t-test) and that the loss of H3K9me3 upon ZNF91-CRISPRi is also significant. The result of this new experiment is inserted into Fig.2d and 2k. It is also inserted below.

We have also designed and used an additional upstream primer pair. The results from this primer pair are very similar to the results obtained with the previous pair. This additional analysis confirms our findings. The new data is inserted as Extended Data Fig.2d&h.

We have also designed and tested several primer pairs targeting the downstream XDP-SVA junction. Unfortunately, none of these primers work. It seems as if this genomic region is not compatible with the design of qPCR-primers. Thus, we are limited to analyse the upstream region of the XDP-SVA.

Reviewer #3:

Remarks to the Author:

This study aims to dissect the mechanisms that control the regulatory impact of SVA retrotransposons in the human genome, focussing on the interesting case of an intronic SVA insertion in the TAF1 gene that causes X-linked dystonia-parkinsonism (XDP). They conduct a series of experiments using ips cells and derived neural progenitor cells (NPCs) to show that, in most cases, both H3K9me3 (controlled by ZNF91) and DNA methylation are required to repress general SVA expression and their regulatory effects on nearby genes. In the case of the SVA in the intron of TAF1, they find that loss of these repressive marks causes a higher degree of partial intron retention and lower level of expression of a downstream exon, indicating a more pronounced molecular defect. They then identify young, insertionally polymorphic SVA elements in their cell lines and examine transcriptional effects when H3K9me3 and DNA methylation are knocked down. The paper was well written with appropriate references and the experiments were generally clear and well controlled. The quality of the data is high and the approaches and statistics valid.

We would like to thank the reviewer for the overall positive comments on our manuscript. We have now generated a new version of the manuscript with revisions in line with the suggestions from the reviewers. We genuinely feel that the manuscript is greatly improved. Changes in the manuscript are outlined in the point-by-point response below and can be found in red font in the new version of the manuscript.

Comments:

1. The fact that most SVAs are heavily methylated, bind ZNF91 and are marked with H3K9me3 has been reported before, so the main novelty or originality of this study is demonstrating the effect on expression of TAF1 when the repressive marks on the SVA are removed, with the results shown in figures 5e and 5d. This is an interesting finding, and it may help explain the late age of onset of XDP since, as the authors mention, it is known that DNA methylation levels decrease in the aging brain. However, it was

somewhat disappointing that no further insight was gained into the actual mechanism by which the SVA causes the transcriptional defect of TAF1 in the first place.

We agree with the reviewer that it is very important to find out why the SVA insertion causes the transcriptional defect of *TAF1*. However, this is a very challenging problem to solve. What is important and novel about our observations is that they demonstrate that chromatin modifications, such as DNA methylation and H3K9me3, over the SVA are important for the magnitude of effect the SVA pose on *TAF1* expression. Thus, the molecular pathology of XDP is now linked to the chromatin status of the XDP-SVA. This is novel information that provides additional mechanistic insights into the molecular pathology of XDP and opens up for new avenues of research in the future.

2. In figure 1f, please show the location of the SVA element in the genome browser tracks and indicate its transcriptional orientation with respect to the TAF1 gene. A quick perusal of the 2018 Cell paper describing this mutation found that the spurious intronic TAF1 transcription terminated 5' to the SVA. It should be made clear if this present study also found the same thing. Is this SVA full length with an intact promoter? Is there any evidence of transcription initiating or terminating in that SVA when H3K9me3 and DNA methylation are removed? Perhaps I missed it, but I could not find mention of this in the manuscript.

In the new version of the manuscript, we have clarified the location of the SVA in the TAF1 gene (Fig.1f). We have also investigated where the intron retention ends. Our observations are in line with the 2018 Cell paper. The intron retention terminates at the 3' end of the SVA. We also find that the intron retention ends at the 3' end of the SVA in ZNF91&DNMT1-CRISPRi-NPCs, in which where the intron retention is greatly increased. However, it should be noted that transcripts coming from the actual XDP-SVA sequence is severely limited due to mappability issues. Thus, short-read sequencing does not allow to exactly pinpoint where the transcription ends. Using other long-read sequencing approaches to study alternative transcripts would be needed to clarify the precise transcript (see e.g <https://doi.org/10.1101/2022.10.21.513169>). However, we believe this lies well outside the scope of our study.

When it comes to antisense transcripts that originate from the XDP-SVA, we find no evidence for this in Ctrl-NPCs. However, in the ZNF91&DNMT1-CRISPRi NPCs we find some evidence for such transcripts but as we previously noted, we are severely limited by the poor mappability of the XDP-SVA. We have addressed this issue using CUT&RUN for H3K4me3. The H3K4me3 CUT&RUN analysis supports

that the XDP-SVA becomes transcriptionally active in ZNF91&DNMT1-CRISPRi NPCs (see also the response to comment #5 below).

In the new version of the manuscript, we have inserted a new supplementary figure that illustrates the termination of the TAF1-transcript in relation to the XDP-SVA and the presence of antisense transcripts. The data is inserted into Extended Data Fig.1b and mentioned in the text on p.6 & 21. This figure is also inserted below. The H3K4me3 analysis is inserted as Extended Data Fig.3e.

b, Genome browser tracks showing RNA-seq, H3K9me3 and H3K4me3 in the *TAF1* gene in control, ZNF91-CRISPRi, ZNF91&DNMT1 CRISPRi conditions in CNPC1 and XNPC1. The reads were mapped to the custom genome, where the XDP-SVA is inserted (See Methods). The XDP-SVA is depicted in red.

3. In the latter section of the paper, the authors aim to examine the epigenetic marks on other polymorphic (recent) SVA insertions in two of their cell lines. They state (on page 21): “We identified 22 high confidence polymorphic insertions of the SVA-E and -F subfamilies (average 2.5 kbp in length, range 1.23.8 kbp), of which 14 were shared between the two genomes (Fig.6b). Notably, several of these polymorphic SVAs, which

represent recent TE insertions into the germline of these two individuals, displayed clear hallmarks of local heterochromatinization, including the presence of DNA methylation and H3K9me3.” Then they show two examples in Figures 6c and d. Are these just the most exceptional cases? For how many of the 22 insertions were they able to identify marks of heterochromatinization? The authors should give some indication of the generality of their findings.

We agree with the reviewer (as well as Reviewer #1) that this is an important point. We have now collected this information into a new supplementary table. This table includes information on the location and size of the SVA insertion as well as the DNA methylation level. It also includes information on if the SVA is flanked by H3K9me3 or H3K4me3 and if these marks are dependent on ZNF91.

Lastly, it contains information on detected transcripts originating from the SVA upon ZNF91&DNMT1-CRISPRi. All this information is found in the new Supplementary Table 1. We mention this new table in the text on p.23 and 30.

4. In Figure 6c, the RNA-seq tracks are confusing until one realizes that the EMC4 gene is oriented opposite to SLC12A6. Please make this clear in the figure. Similarly, in Figure 6d, the top part of the figure seems to show ATP5PF and GABPA oriented in the same direction but in fact these two genes are oriented in opposite directions and appear to be using a bidirectional promoter (according to Genome Browsers). Once this is clarified, the stranded RNA-seq tracks will be easier to understand.

We have now generated a new version of Fig 6 where the orientation of the genes and the SVA insertions are clarified. The new versions are also inserted below.

5. For these same two examples, the RNA-seq seems to indicate that the SVAs are being transcribed when the repressive epigenetic marks are removed but the authors show no other data to support this. Are these SVAs actually close to full length with structurally intact promoters? Were these the only two examples where evidence of polymorphic SVA transcription upon release from epigenetic repression was observed?

We agree with the reviewer that this is an important point. In fact, there is an ongoing debate in the field regarding the SVA promoter. It is still unclear exactly where the SVA promoter is located and there is even data suggesting that SVA expression is mostly passive because of readthrough expression. A key challenge

here is that SVA-transcripts, in particular those coming from polymorphic SVAs, are impossible to map with short reads making it challenging to determine where transcription starts.

To address this issue, we have now performed CUT&RUN analysis for the histone mark H3K4me3, which is associated with active promoters. Since the signal of this histone modification spreads to the unique flanking genomic context, this approach allows for an accurate identification of transcriptionally active individual transposon loci (PMID: 37910626). We performed this experiment in control NPCs and ZNF91&DNMT1-CRISPRi-NPCs (both XDP and control lines). Our data demonstrates that most full-length SVAs gain H3K4me3 upon ZNF91&DNMT1-CRISPRi, including several of the polymorphic SVAs. This analysis confirms that SVAs become transcriptionally active when DNA-methylation and H3K9me3 is lost and suggest that the SVA promoter is driving expression. This new data, which provides novel insights into SVA transcription, is inserted into Fig.4e,g, Fig.5b, Fig 6c&d and Extended Data Fig.3d.

When it comes to the XDP-SVA in the TAF gene we used the qPCR-CUT&RUN method to investigate the presence of H3K4me3. These results demonstrated that also the XDP-SVA becomes transcriptionally activated in ZNF91&DNMT1-CRISPRi NPCs. This new data is inserted into the new Extended Data Fig3e. The figure is also inserted below.

6. It has been reported in the literature that length of a (CCCTCT)_n repeat within the SVA element is an age-at-onset modifier in XDP. Perhaps it would be worth mentioning if the cell line models used in this study harbor repeat sizes typically found in patients.

We do not have access to information on the length of the hexameric repeat when the samples used for iPSC generation were sampled. However, the age of onset of these patient are in the normal range, which indicate a hexameric repeat length of around 35-45 repeats. However, we have now been able to estimate the repeat length in the cell lines where we have ONT data. We find that the repeat lengths are 24 and 25 repeats in the XNES1 and XNES3 lines. This information has been added in the manuscript to Table 1. It should be noted that this repeat length is a bit shorter than expected, raising the possibility that the hexameric repeat is unstable in cell lines upon passaging. Future work is needed to resolve this question. Still, 24 and 25 repeats represent a long hexameric region, which is a hallmark of the XDP-SVA.

Decision Letter, first revision:

Message: Our ref: NSMB-A48559A

12th Mar 2024

Dear Professor Jakobsson,

Thank you for submitting your revised manuscript "Mini-heterochromatin domains constrain the cis-regulatory impact of SVA transposons in human brain development and disease" (NSMB-A48559A). It has now been seen by the original referees and their comments are below. The reviewers find that the paper has further improved in revision, and therefore we are happy to accept it in principle in Nature Structural & Molecular Biology, pending minor revisions to satisfy the referees' final requests and to comply with our editorial and formatting guidelines.

We are now performing detailed checks on your paper and will send you a checklist detailing our editorial and formatting requirements in about two weeks. Please do not upload the final materials and make any revisions until you receive this additional information from us.

Sincerely,

Dimitris Typas
Associate Editor
Nature Structural & Molecular Biology

ORCID: 0000-0002-8737-1319

Reviewer #1 (Remarks to the Author):

The authors have answered all my questions and suggestions.

Reviewer #2 (Remarks to the Author):

the authors addressed my concerns. this is a nice paper addressing the separable and important functions of ZNF91 and Dnmts in regulating the SVA at the TAF1 locus and playing a role in XDP pathology

Reviewer #3 (Remarks to the Author):

The authors have done a very good job of responding to my comments and those of the other reviewers. They have added additional data and discussion which improve the manuscript. I have no further major concerns.

Final Decision Letter:

Message: 17th Apr 2024

Dear Professor Jakobsson,

We are now happy to accept your revised paper "Mini-heterochromatin domains constrain the cis-regulatory impact of SVA transposons in human brain development and disease" for publication as an Article in Nature Structural & Molecular Biology.

You will not receive your proofs until the publishing agreement has been received through

our system.

Your paper will be published online soon after we receive proof corrections and will appear in print in the next available issue. You can find out your date of online publication by contacting the production team shortly after sending your proof corrections.

If you have not already done so, we strongly recommend that you upload the step-by-step protocols used in this manuscript to the Protocol Exchange. Protocol Exchange is an open online resource that allows researchers to share their detailed experimental know-how. All uploaded protocols are made freely available, assigned DOIs for ease of citation and fully searchable through nature.com. Protocols can be linked to any publications in which they are used and will be linked to from your article. You can also establish a dedicated page to collect all your lab Protocols. By uploading your Protocols to Protocol Exchange, you are enabling researchers to more readily reproduce or adapt the methodology you use, as well as increasing the visibility of your protocols and papers. Upload your Protocols at www.nature.com/protocolexchange/. Further information can be found at

www.nature.com/protocolexchange/about.

Please note that *Nature Structural & Molecular Biology* is a Transformative Journal (TJ). Authors may publish their research with us through the traditional subscription access route or make their paper immediately open access through payment of an article-processing charge (APC). Authors will not be required to make a final decision about access to their article until it has been accepted. Find out more about Transformative Journals

Sincerely,

Dimitris Typas
Associate Editor
Nature Structural & Molecular Biology
ORCID: 0000-0002-8737-1319